# A post-Brexit intergroup contact intervention reduces affective polarization between Leavers and Remainers short-term
Nicole Tausch [1] ✉, Michèle D. Birtel [2], Paulina Górska[3], Sidney Bode[1] & Carolina Rocha[4]

With mounting evidence of the harmful societal consequences of affective polarization, it is crucial to find ways of addressing it. Employing a randomized controlled trial, this study tested the effectiveness of an intervention based on theories of intergroup contact and interpersonal communication in reducing affective polarization in the context of Brexit. Participants were 120 UK self-identified Leavers and Remainers. Sixty Leaver-Remainer dyads were randomized to engage in either a facilitated intergroup interaction or a control interaction, which was equivalent in structure and tone but was unrelated to Brexit identities. Different aspects of affective polarization were assessed one month prior, immediately after, and one month after the intervention. Results indicate that the intervention increased warmth toward the outgroup, reduced unfavourable attributions of the sources of outgroup positions, and increased willingness to compromise, but only short-term. There were no statistically significant longer-term effects of the intervention. Evidence of selective attrition further suggests that those with more extreme baseline opinions were more likely to drop out. Our findings highlight the challenges of designing effective interventions that engender enduring attitude change in polarized contexts and of engaging those with extreme political views. This study can provide a useful framework for future research.

Disagreement and debate over moral and political issues, such as abortion, environmental law, or immigration, are part and parcel of functioning democratic communities[1]. Political conflicts can, however, also create stark group boundaries and result in affective polarization, whereby opposing sides increasingly view each other with hostility and disdain[2–4]. Affective polarization is evident in democracies world-wide[5–7], and has widespread societal and political consequences[8–10]. It predicts discrimination against outgroup members in economic[11] and employment contexts[3,12] and avoidance and separation between groups in a range of life domains[2,13–15]. There are also concerns that affective polarization erodes commitment to democratic principles; it reduces willingness to compromise with the other group[16,17], increases acceptance of norm violations by ingroup candidates[18], and predicts support for political violence[19]. Some scholars have therefore argued that affective polarization jeopardizes democracy itself and underscored the urgency of finding ways of reducing it[5,20].

Can interventions informed by psychological science counteract affective polarization? Building on insights from the intergroup contact and communication literatures, we designed a multi-faceted intervention that brought together UK Leavers and Remainers, two groups that have been embroiled in a divisive debate over Britain's exit from the European Union ("Brexit") that has polarized the country[21] and generated significant mutual hostility between Leavers and Remainers[22]. We evaluate the effectiveness of our intervention in ameliorating different aspects of affective polarization in this context, and in increasing willingness to compromise with the opposing group. Our research has the potential to provide evidence-based guidance to policy makers, community groups, and citizen initiatives working to create platforms that facilitate dialogue and cooperation across political divides.

A vast literature on intergroup contact suggests that contact might be beneficial in reducing negative intergroup attitudes, especially when it takes place under conditions of equal status, involves cooperation in pursuit of a

[1]University of St Andrews, School of Psychology and Neuroscience, St Andrews, UK. [2]University of Greenwich, Institute for Lifecourse Development, School of Human Sciences, London, UK. [3]University of Warsaw, Faculty of Psychology, Warsaw, Poland. [4]University of Dundee, School of Business, Dundee, UK. ✉e-mail: nt20@st-andrews.ac.uk

common goal, and is supported by institutional or informal norms[23–25]. Contact's effects are thought to be enhanced to the extent that it allows participants to become personally acquainted[26] and encourages self-disclosure[27,28] and generates positive affective ties to the outgroup[29]. Nonetheless, much of the research underpinning contact theory has been correlational[30] and recent longitudinal work has yielded mixed findings regarding the within-person effects of contact[31–33]. Importantly, there is a dearth of high-quality experimental intervention studies and the few studies available indicate that the effectiveness of contact interventions varies substantially according to the type of prejudice addressed[30].

Research on contact in the political domain has frequently focused on relations between Republicans and Democrats in the United States. It suggests that contact across political divides might have a range of positive consequences. Self-reported high-quality contact predicts reduced outparty prejudice[34,35] and people with more (as opposed to less) politically diverse social networks tend to show greater tolerance for opposing viewpoints and support for democratic values[36,37]. Initial experimental work assessing the effects of actual interactions on affective polarization similarly points to a number of benefits, but is inconclusive about the causal effect of intergroup contact[38–40], or leaves unresolved questions about how intergroup differences should be addressed for contact to be effective[41]. By applying principles of both intergroup contact and dialogue, and by conducting a randomized controlled trial with both immediate and delayed outcome assessment, the current research provides a causal test of the impact of a contact intervention in the political domain. It thereby responds to calls for more rigorous, policy-relevant research on the effects of intergroup contact more generally[30].

Our research is conducted in the context of polarization around Brexit, where opposing political identities and intergroup tensions emerged over a relatively short period of time[22]. On 23rd June 2016, 52% of the UK electorate voted in favour of Britain leaving the European Union. This unexpected outcome instigated a tumultuous period in British politics that led to an extended deadlock in parliament over the Brexit deal, an erosion of public trust in political institutions[42], and divisive party politics ever since. The "Leave" vs "Remain" campaigns ahead of the referendum also gave rise to a novel political identity, with a substantial proportion of the British public strongly identifying as either Leaver or Remainer[43]. These identities have generated negative stereotypes and evaluative biases consistent with affective polarization[22] and have negatively impacted both public civility and personal relationships[42,44].

Research addressing potential interventions that reduce the division between Leavers and Remainers is lacking. This study aims to fill this gap by testing the effects of a structured, facilitated one-off intergroup interaction between Leavers and Remainers. To maximize its potential to reduce affective polarization, we considered two main issues: First, for contact to have positive effects that generalize to the outgroup as a whole, group identities need to be made salient without generating anxiety that could undermine the quality of contact. Approaches to intergroup contact have variably underlined the importance of deemphasizing group boundaries during contact to lessen anxiety and avoid stereotyping (de-categorization)[45,46], of making categories salient during contact to aid generalization of positive affect (categorization)[47], and of creating a superordinate category that incorporates both groups and thereby reduces bias (re-categorization)[48]. Our intervention combines these approaches in a temporal sequence such that personalized contact, which de-emphasizes group memberships, dampens anxiety and leads to interpersonal liking, happens first; group memberships are made salient subsequently to achieve generalization of positive affect to the outgroup as a whole; and recategorization is encouraged at a later stage to achieve maximum prejudice reduction[49].

Second, intergroup contact should allow participants to engage with the outgroup in a meaningful way. Contact theory has largely neglected the content of intergroup interactions[50] and offers little guidance on how to best approach substantive disagreements between groups. There is, however, often a strong desire among participants in intergroup encounters to discuss

actual group differences[51] and avoiding such topics can result in cynicism[52] and limit contact's effects on more consequential outcomes such as willingness to make concessions over policy issues[41,50]. Debating disagreement directly is, however, likely to be counter-productive as it undermines the quality of contact and may further polarize groups[50,53]. To address these issues, our intervention incorporates insights from work on the communication principles of dialogue[54]. Dialogue contrasts from debate, which is aimed at winning an argument, or discussion, which focuses on persuading others[55]. By encouraging participants to listen actively and validate each other's perspectives, it enables participants to exchange views in a non-adversarial way and provides opportunities to discover common ground[56,57]. The potential effectiveness of this approach in the context of intergroup conflict is supported by research demonstrating the benefits of expressing one's point of view ("perspective-giving") in intergroup encounters[58,59] and the de-polarizing effects of feeling understood by an outgroup[60,61]. Moreover, those exposed to outgroup viewpoints are more receptive to outgroup arguments if they expect to be treated respectfully[62]. Actual conversational receptiveness, which includes explicit acknowledgement of understanding and positive statements instead of negation, generates trust[63] and elicits higher levels of collaboration intentions[64] and willingness to talk[65].

Drawing on these literatures, we developed a three-stage, facilitated interaction that brought together Leavers and Remainers in an online face-to-face conversation. We began the interaction with the "Fast Friends" procedure, an established technique to encourage self-disclosure and create interpersonal liking (Stage 1)[65]. This was followed by a task which made Brexit categories salient and facilitated respectful dialogue about each other's reasons for supporting either "Leave" or "Remain". Principles of respectful dialogue, which highlighted the importance of acknowledging as opposed to negating one's partner's perspective, were laid out at the beginning of each interaction by the facilitator, repeated prior to discussing political issues at the beginning of Stage 2, and emailed to participants in advance of the interaction (Stage 0). This created a clear norm supporting respect and active listening throughout the interaction. Stage 3 highlighted a superordinate goal (protecting mental health during the pandemic) and asked participants to brainstorm cooperatively to find solutions. This task was included to make salient a shared, superordinate identity[48,66] to further facilitate prejudice-reduction and to finish the conversation in a positive manner. Table 1 gives an overview of the stages of the intervention and summarizes supporting literature.

We evaluate the effects of this intervention on a spectrum of variables capturing different aspects of intergroup affect and cognition. In line with previous work on affective polarization[2,10,22] we assess changes in negative affect towards, and stereotypes of, the outgroup. We measure both general outgroup warmth in the form of a feeling thermometer as well as specific negative emotions toward the outgroup, which are key drivers of negative intergroup behaviour[67,68]. We include trait ratings on the two main

**Table 1 | Intervention elements and supporting literature**

| | Content | Supporting literature |
|---|---|---|
| Stage 0 | Ground rules of respectful dialogue | Equal status contact[24]<br>Supportive social norms[24]<br>Equality-based respect[62]<br>Intergroup dialogue[56,57] |
| Stage 1 | Personal acquaintance task ("Fast Friends") | Acquaintance potential[26]<br>Decategorization[45,46]<br>Cross-group friendship[27,29]<br>Self-disclosure[27,28]<br>Interpersonal closeness[65] |
| Stage 2 | Respectful intergroup dialogue | Salient categorization[47]<br>Intergroup dialogue[56,57]<br>Perspective-giving[58,59]<br>Conversational receptiveness[63,64] |
| Stage 3 | Cooperative contact toward a superordinate goal | Cooperation toward a common goal[24]<br>Recategorization[48] |

**Fig. 1 | Overview of study design.** Timeline of the three measurement points (T1, T2, and T3) and intervention points. Means (M) and stanadard deviations (SD) of distance between points between measurement points.

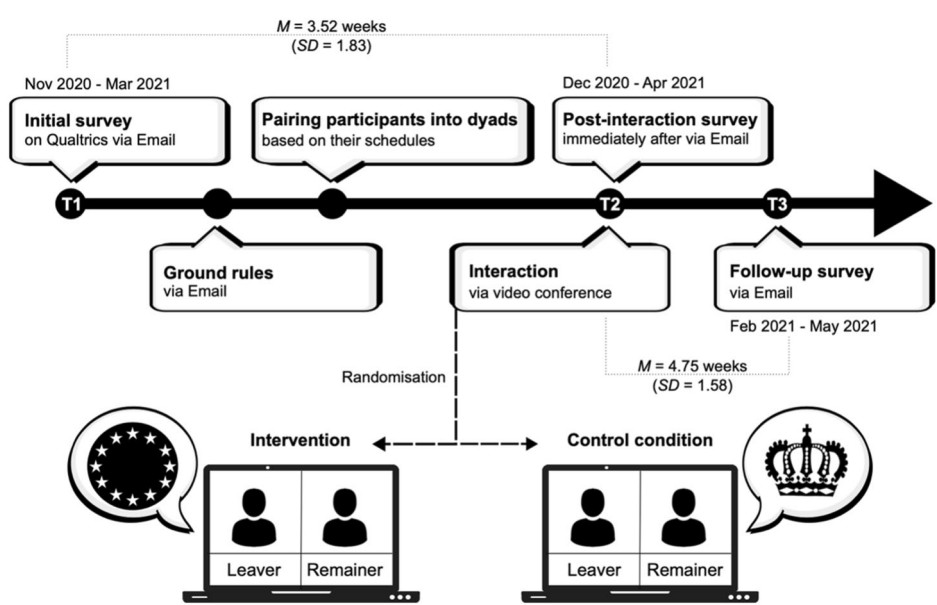

dimensions of social perception[69], competence ('intelligent') and warmth ('intolerant').

We further consider the impact of contact on meta-beliefs; that is, beliefs about how the other group thinks about and feels toward the ingroup[70]. Negative meta-beliefs among political opponents tend to be widely exaggerated, with people overestimating the extent to which the outgroup views the ingroup with hostility[71–73]. This can have harmful consequences and result in avoidance and defensive behaviour that violates community standards[71]. Interventions that correct meta-perceptions, either through information[72,73] or contact[74,75], can, in turn, reduce intergroup conflict and increase willingness to engage with the outgroup. We assess both meta-stereotypes ('intelligent' and 'intolerant') and negative meta-emotions[76].

We also assess several outcomes that have received less attention. Specifically, we included measures that tap into (mis-)perceptions about the reasons that underlie the outgroup's positions[34], as well as the emotion of contempt, which has been linked to incivility in political discourse[77,78] and to readiness to support violent collective action[79]. Avoidance of contact with political opponents and immersion in a highly selective media environment means that people may have a limited understanding of the rationales and moral concerns underlying oppositional viewpoints[80]. We examine three central attributional dimensions that are subject to group biases: externality (i.e., the extent to which attitudes are influenced by external sources such as the media), emotionality (e.g., the degree to which attitudes are influenced by emotions), and rationality (e.g., the extent to which attitudes are formed through rational thought and contemplation)[34]. People tend to attribute ingroup positions to a greater extent to rationality and to a lesser extent to emotionality or externality, compared to outgroup positions[81]. This has implications for how people relate to those with opposing views. Negatively distorted perceptions of the political outgroup as misguided and irrational might result in views that the outgroup is intellectually or morally inferior, and unlikely to be influenced through rational debate. Appraisals of low competence, bad character and lack of influenceability, in turn, elicit contempt[68,69,82], a particularly damaging emotion which has been shown to precede the breakdown of relationships[83], reduce reconciliatory intentions[82], and to motivate physical and psychological distancing[82].

To assess the extent to which our intervention impacts on outcomes that have implications for political behaviour and democratic processes, we measure participants' willingness to cooperate and compromise with the outgroup over political issues, their ideological position on Brexit, as well as their willingness to engage in collective action on behalf of their ingroup. We explore the idea that contact and exposure to outgroup positions might moderate Brexit-related opinions and reduce the impetus to engage in collective action to defend the ingroups interests[84,85]. Finally, we tested our intervention's impact on traditional outcomes examined in the contact literature[86], specifically anxiety about interacting with the outgroup, empathic concern for the outgroup, and willingness to take the outgroup's perspective.

We measured these outcomes at three time points; approximately one month prior to the intervention to obtain baseline positions, immediately after the intervention, and approximately one month after the intervention to assess whether any effects are sustained longer-term. To isolate the effects of intergroup contact on our outcome variables, we compared our intervention to the effects of a control interaction, which was equivalent in structure and tone, but which did not make Brexit identities salient. This ensured that any observable effects on our outcome variables cannot be attributed to having participated in a positive interaction per se. Figure 1 summarizes the timeline of events and measurement points for the study.

We hypothesized that, compared to the control group, participants in the intervention condition would exhibit greater reductions in negative affect, negative stereotypes, negative meta-perceptions, unfavourable attributions and contempt, and increased support for compromise with the outgroup. We also expected reductions in intergroup anxiety and increases in empathic concern and perspective-taking. Finally, we predicted reductions in collective action tendencies and explored whether contact moderated beliefs around Brexit (i.e., whether it engenders lower Brexit support for Leavers and increased support among Remainers). We also explored differences between Leavers and Remainers for the predicted effects. Remainers, who lost the vote, show higher levels of negativity toward Leavers than vice versa[22] and were twice as likely as Leavers to break off relationships in the aftermath of Brexit[87]. Moreover, polling data indicates that Remainers misperceive Leavers reasons for supporting Brexit to a greater extent than Leavers misperceive Remainers reasons for voting against[88]. It is therefore possible that Remainers exhibit greater changes on our outcome measures than Leavers[89].

## Methods

Although not pre-registered, analyses were conducted after data collection was complete, and all outcome measures and conditions are reported. We also included several relevant background variables at T1 (e.g., identification as Leaver/Remainer, prior outgroup contact) as well as exploratory variables not further discussed here.

The final sample size was determined by practical and budgetary constraints. All relevant ethical regulations were followed and informed consent was obtained in accordance with the University Teaching and Research Ethics Committee at the University of St Andrews.

## Recruitment and sample

Data were collected between November 2020 and May 2021, a period during which Brexit continued to be a salient issue in public discourse and Brexit identities remained highly important among the British population[90]. Participants were invited to take part in a research project on social relations in post-Brexit Britain through social media adverts and Prolific Academic (www.prolific.co). They were informed that their participation would involve the completion of three online surveys about their views related to Brexit as well as participation in an online face-to-face discussion with another participant, and that, to be eligible to participate, they had to currently identify as either Leaver or Remainer. Participants were informed that the interaction would involve discussing a current UK-wide issue (Brexit or the future of the Royal family) and that it may involve interacting with someone who holds an opposing view on the issue. Respondents who expressed interest were provided with a detailed information sheet that outlined the general study procedure, highlighted that their participation might entail interacting with another participant who disagrees with them on the discussion topic, and reminded them of their right to withdraw from the study. All participants consented to take part and confirmed that they were over the age of 18. We obtained an initial sample of 586 respondents who completed the T1 (baseline) survey. Of these respondents, 484 (82.6%) indicated that they would like to continue with the study. These participants were contacted by the research assistants via the Prolific messaging system to indicate their availability for the available timeslots for the interaction. Of these participants, $N = 172$ never responded and of the $N = 310$ participants who did initially respond, $N = 90$ never confirmed a timeslot or, in a few cases, did not show for a scheduled interaction. Finally, $N = 102$ participants were Remainers who could not be assigned an interaction partner since many more Remainers ($N = 394$) than Leavers ($N = 192$) entered the study to begin with (see further details on participant recruitment in the Supplementary Methods 1). As compensation, participants recruited via social media received £20 after completion of Part 2 and £5 after completion of Part 3, in the form of Amazon vouchers. Prolific participants received £2.50 upon completion of the T1 survey, an additional £13.50 upon completion of the interaction and T2 survey, and £5.50 upon completion of the T3 survey via Prolific bonus payments. In addition, all participants had the opportunity to enter into a prize draw for a £100 Amazon voucher upon completion of the study.

A total of 120 participants (60 Leavers and 60 Remainers) completed the interaction. All of these participants completed both the T2 (administered immediately after the interaction) and the T3 survey. Sixty-two participants (31 Leavers and 31 Remainers) were in the experimental group and 58 participants (29 Leavers and 29 Remainers) were in the control group. This slight imbalance was due to no-shows for some of the scheduled interactions. The sample consisted of 67 women and 53 men with an age range of 24–79 ($M_{age} = 44.90$, $SD = 13.19$). Most participants were of white ethnic background ($N = 108$), six were of Black, 3 of Asian, and one of mixed ethnic background. The Supplementary Methods 2 provide further details on sample composition. Data related to outgroup views from one participant who changed their identification from Remainer to Leaver at time 2, and one participant who changed from Leaver to Remainer at time 3, were treated as missing values at T2 and T3, and at T3, respectively. Post-hoc sensitivity power analysis for mixed designs with within-between interactions, with average correlations between repeated measures of outcomes of $r_{T1-T2} = 0.545$ and $r_{T1-T3} = 0.607$ and considering dyads ($N = 60$) as the unit of analysis revealed that our final sample size allows 80% power to detect effect sizes of $d = 0.350$ for tests of short-term effects and $d = 0.326$ for tests of long-term effects. Note that the detectable effects are smaller than the lower-bound of the confidence interval of overall contact effects reported in the largest meta-analysis of contact effects[25]. We therefore believe that this sample size is sufficient to test our main hypotheses. There was only a very small percentage of missing data (≤2.5%) scattered across variables. To deal with this, we used full information maximum likelihood estimation in our main analyses (FIML[91]). FIML estimation produces less biased results compared to more traditional methods to handle missing data such as listwise and pairwise deletion[92].

## Procedure and intervention

Participants first completed an online questionnaire which included a battery of demographic measures, their vote in the 2016 EU referendum, their current orientation on the issue (i.e., self-identification as Leaver or Remainer), their level of identification as either Leaver or Remainer, and their views on the impact of Brexit on several issues. The main part of the questionnaire assessed participants' contact experiences with, and views of, the political outgroup, their meta-perceptions, compromise willingness and collective action intentions. The intervention took place on average 24.67 days ($SD = 12.81$) after completion of the pre-intervention survey. Participants were scheduled to interact in dyads, consisting of one Leaver and one Remainer, via video conference. Dyads were randomly assigned to either the experimental or control group using a random number generator. In the days preceding the interaction, participants were emailed the video link and several ground rules to ensure privacy and confidentiality of the interaction and respectful engagement with the interaction partner (see Supplementary Methods 3).

The interactions were facilitated by one of two trained moderators. Once both participants had joined the online meeting, the moderator greeted the participants, gave an overview of the session structure, and reminded participants of the ground rules and their right to withdraw. The moderator shared their video only during the introduction and conclusion. Regardless of condition, participants first engaged in an interpersonal interaction (Stage 1). Based on the Fast Friends procedure[65] the interaction partners took turns reading and answering questions that progressively encouraged greater self-disclosure (e.g., "What would constitute a perfect day for you?"). In the following section (Stage 2), participants in the experimental condition were asked to exchange their views on Brexit. This involved participants' disclosing their stance on Brexit and explaining the reasons underlying their view. Participants in the control group were asked to exchange their views on a controversial (though somewhat less politically charged) topic: the then recent departure of Prince Harry and Meghan Markle from the British royal family. This topic frequently featured in the media at the time and generated a good amount of engagement and difference in opinion to mimic the style of conversation in the experimental group (see Supplementary Notes 1). Importantly, although dyads in the control condition also consisted of a Leaver and a Remainer, participants in the control condition were not aware that they were interacting with someone with an opposing Brexit identity and this condition therefore did not constitute intergroup contact along the Leaver-Remainer divide[47]. The choice of this control group ensured that any experimental effects could be attributed to engagement in *intergroup* contact rather than a positive interaction per se. Following this stage, participants in both conditions were asked to engage in a cooperative brainstorming task (Stage 3), which required them to identify actions local councils should take to address mental and physical wellbeing during the pandemic. To conclude the session, the moderator thanked participants for their time and provided the link to the T2 survey. These included measures of their current mood, their perception of the interaction and their interaction partner, and repeated measures on their views on Brexit and of the political outgroup. Approximately 30 days after the interaction, participants were sent a final survey which included measures of their levels of contact with the outgroup in the last month, repeat measures of their views on Brexit and of the political outgroup, and additional demographic and background variables. Participants completed the final survey on average 33.26 days ($SD = 11.07$) after the intervention.

## Measures

We only present measures relevant for the current paper here. The complete set of measures, decisions about scale formation, as well as the full list of items for each construct, are presented in the Supplementary Methods 4.

**Demographics and background measures.** Participants indicated their age, gender (female, male, non-binary, prefer to self-describe), highest education level completed (1 = no formal schooling, 2 = primary school, 3 = secondary school, 4 = GCSE or similar, 5 = A-level or similar, 6 = undergraduate education, 7 = postgraduate education), and their country of residence (England, Scotland, Wales, Northern Ireland, other) at T1. At T3, participants also indicated their ethnicity (White, Asian, Black, Arab, Latinx/Hispanic, mixed/multiple ethnicities, other ethnic group). At T1, they were asked how they voted in the EU Referendum, what their current stance on Brexit is (Leaver, Remainer), and their *level of identification* as Leaver or Remainer on a five-item scale (e.g., "Being a Leaver/Remainer is an important part of how I see myself", α = 0.84). *Prior contact* was measured by asking participants how many of their acquaintances, close friends, and family members are Remainers/Leavers [outgroup members] (1 = none, 2 = a few, 3 = about half, 4 = more than half, 5 = most, 6 = all). We combined responses into an overall index of social network penetration (α = 0.80). We also asked them how frequently they (knowingly) interacted with outgroup members over the last few months (1 = never, 2 = rarely, 3 = occasionally, 4 = often, 5 = very often). Prior *contact quality* was captured using a 5-item scale. Participants rated the extent to which their contact with outgroup members was overall respectful, friendly, comfortable, and equal on 7-pointy Likert-scales. The items were combined into a composite of contact quality (α = 0.85).

**Ratings of the interaction.** To assess the equivalence of the two conditions in terms of experience we assessed participants' post-interaction affect, the perceived quality of the interaction, and experienced respect during the interaction, and the perceived similarity to the interaction partner. Directly following the interaction, we measured participants' *state affect* using the Positive and Negative Affect Schedule[93]. On a 5-point Likert scale ranging from 1 = not at all to 5 = very much, participants indicated the extent to which they felt a range of positive and negative emotions (e.g., inspired, upset). Factor analysis extracted two separate factors and the items were combined to form indices of positive (α = 0.84) and negative (α = 0.67) state affect. The perceived *quality of interaction* was measured using the same 5 items as above and combined into an index of quality of the interaction (α = 0.85). We assessed *equality-based respect experienced during the interaction* with six items adapted from previous work[94] (e.g., "My interaction partner communicated with me as with a person of equal worth", "My interaction partner treated me as a decent person"; α = 0.90). Finally, participants were asked to rate the extent to which they thought that there are *similarities* or differences between them and their interaction partner in terms of: values, everyday concerns, and strength of community ties (α = 0.73).

**Outcome measures.** The focal outcome variables were assessed at all three time points. *Warmth towards the outgroup* was measured on a feeling thermometer ranging from 0 to 100. Responses were given on a slider with the following instruction: How do you feel toward [outgroup members]? Please rate [outgroup members] on a thermometer that runs from zero (0) to a hundred (100) degrees. The higher the number, the warmer or more favourable you feel towards [outgroup members]. The lower the number, the colder or less favourable you feel. If you feel neither warm nor cold towards [outgroup members], rate them at 50.

To assess *negative outgroup emotions*, participants indicated to what extent they felt anger, disgust, contempt, respect (R) toward [outgroup members] on scales ranging from 1 (*not at all*) to 7 (*extremely*). The 'respect' item was reversed, and all items were then combined into reliable indices of negative emotions toward the outgroup ($α_{T1} = 0.79$; $α_{T2} = 0.81$; $α_{T3} = 0.79$).

Outgroup stereotypes were measured by asking participants: "To what extent do you think [outgroup members] possess the following traits: intelligent, intolerant". Answer scales ranged from 1 (*not at all*) to 7 (*very much*). As these items were only moderately correlated, we analysed them separately.

To assess *meta-emotions* participants were asked to answer the following questions on scales ranging from 1 (not at all) to 7 (very much): "Based on your experience and observations, to what extent do you think [outgroup members] feel the following emotions towards [ingroup members]: anger, disgust, contempt, respect (R)"[76]. The items were combined to form a reliable index of negative meta-emotions ($α_{T1} = 0.84$; $α_{T2} = 0.90$; $α_{T3} = 0.85$). Similarly, to measure *meta-stereotypes*, participants used scales ranging from 1 (*not at all*) to 7 (*very much*) to answer the following two questions: "Based on your experience and observations, to what extent do you think [outgroup members] rate [ingroup members] on the following traits:" intelligent, intolerant. As the two items were only moderately correlated, we analysed them separately.

We adapted six items from previous work[34] to assess *attributions about the sources* of the outgroup's political views in terms of *rationality* (e.g., "They have rational arguments to support their ideas"), *emotionality* (e.g., "They hold their views because of their emotions"), and *externality* (e.g., "They hold their views because of the influence of media (TV, newspapers, etc.)") on 7-point scales (1 = *does not apply at all*, 7 = *applies very much*). The two items assessing each dimension were combined into indices of rationality ($r_{T1} = 0.51$; $r_{T2} = 0.48$; $r_{T3} = 0.49$), emotionality ($r_{T1} = 0.85$; $r_{T2} = 0.89$; $r_{T3} = 0.82$), and externality ($r_{T1} = 0.38$; $r_{T2} = 0.72$; $r_{T3} = 0.53$).

Based on the theoretical literature on contempt[82], we generated items that assess the main cognitive appraisals and motivational goals of *outgroup contempt* in this context. The final scale consisted of 4 items capturing blameworthiness ("Leavers/Remainers [outgroup] are to blame for the current political problems in this country"), inferiority ("Leavers/Remainers have given me reasons to look down on them") and avoidance ("I currently feel like I would go to some effort to avoid contact with Leavers/Remainers."). The items were combined into reliable indices of contempt ($α_{T1} = 0.82$; $α_{T2} = 0.84$; $α_{T3} = 0.84$).

We adapted three items from the Intergroup Compromise Inventory[95] to measure *willingness to compromise with the outgroup* in this context (e.g., "If [outgroup members'] solution is proven to be better for the country, we should support them") on scales ranging from 1 (*strongly disagree*) to 7 (*strongly agree*). The items were combined into a reliable measure of compromise willingness ($α_{T1} = 0.71$; $α_{T2} = 0.80$; $α_{T3} = 0.81$).

To measure *collective action intentions*, we asked participants how likely they are to engage in three actions (sign a petition, attend a demonstration, write a letter to the government) to protect the interests of Leavers/Remainer [ingroup] on scales ranging from 1 (very unlikely) to 7 (very likely). The items were averaged into a composite index of collective action intentions ($α_{T1} = 0.85$, $α_{T2} = 0.89$, $α_{T3} = 0.89$).

We assessed participants' *Brexit opinions* by asking them to indicate their agreement with 11 items capturing some of the main points of contention in the Brexit debate (e.g., "Brexit will allow the UK to take back control over immigration and its borders"; "Brexit will have a negative impact on the economy, jobs, and prices"), using scales ranging from 1 (*strongly disagree*) to 7 (*strongly agree*). We reversed pro-Brexit items and combined all items into a composite with higher values indicating stronger anti-Brexit views ($α_{T1} = 0.95$; $α_{T2} = 0.94$; $α_{T3} = 0.95$).

Finally, we included items from standard scales in the contact literature to assess intergroup anxiety, empathic concern and perspective taking. We measured *intergroup anxiety* by asking participants to what extent they would feel anxious, comfortable (R), secure (R), and tense if they were the only [ingroup member] interacting with a group of [outgroup members], on scales ranging from 1 (not at all) to 7 (very much). The items were reversed as appropriate and combined to form an index of intergroup anxiety ($α_{T1} = 0.90$; $α_{T2} = 0.92$; $α_{T3} = 0.85$). *Empathic concern for the outgroup* was measured with three items adapted from the empathic concern subscale of the interpersonal reactivity index[96] (e.g., "If I saw a Leaver/Remainer

[outgroup] being treated unfairly, I wouldn't feel much pity for them" (R)). Agreement with the statements was measured on scales ranging from 1 (*strongly disagree*) to 7 (*strongly agree*) and the items were reversed as appropriate and combined to create an index of empathic concern for the outgroup ($\alpha_{T1}$ = 0.76; $\alpha_{T2}$ = 0.77; $\alpha_{T3}$ = 0.72).

Three items assessing *willingness to take the outgroup's perspective* were adapted from the perspective taking subscale of the interpersonal reactivity index[96] ("I sometimes find it difficult to see things from a [outgroups]'s point of view" [R]; "I sometimes try to understand [outgroup members] better by imagining how things look from their perspective"; "When facing political disagreements where I am sure I'm in the right, I believe listening to a [outgroup members'] arguments is a waste of my time" on scales ranging from 1 (strongly disagree) to 7 (strongly agree). Scale reliability analysis indicated that the three items did not form a reliable scale ($\alpha_{T1}$ = 0.40, $\alpha_{T2}$ = 0.53, $\alpha_{T3}$ = 0.49) due to the low item-total correlation of the positively worded item. We therefore dropped this item and reversed and combined the remaining two items into an index of outgroup perspective taking ($r_{T1}$ = 0.35, $p < 0.001$; $r_{T2}$ = 0.31, $p < 0.001$; $r_{T3}$ = 0.33, $p < 0.001$).

### Statistics and reproducibility

We use a type I error rate of 0.05 and report two-tailed significance levels for all analyses. Data distribution was assumed to be normal, but this was not formally tested. Given that our data were nested, we employed multi-level analyses[97] using the MLF estimator in Mplus[98]. A series of three-level models in Mplus accounted for nesting of measurement points (level 1) within individuals (level 2) and nesting of individuals within dyads (level 3). In each of these models, time was coded with a dummy variable comparing measurement at T1 with measurement at T2 (to assess short-term effects) or measurement at T1 with measurement at T3 (to assess longer-term effects). Changes over time are represented by the effects of dummy variables capturing short-term (T1-T2) and longer-term (T1–T3) effects, respectively. Experimental effects are indicated by a significant cross-level interaction between condition (0 = control group, 1 = intervention group), which was specified as a level-3 variable, and the slope representing time. We also

explored whether short- and longer-term effects were moderated by group membership by testing for three-way interactions between time, condition, and group (Leaver = 0 vs. Remainer = 1).

### Reporting summary

Further information on research design is available in the Nature Portfolio Reporting Summary linked to this article.

## Results

### Preliminary analyses of baseline data

**Differences between Leavers and Remainers.** A total of 586 participants ($N$ = 192 self-identified Leavers and $N$ = 394 self-identified Remainers) completed the baseline survey. Comparisons between Leavers and Remainers (see Supplementary Results 1) revealed significant differences on almost all variables: Remainers were more highly identified with their group than were Leavers and had lower levels of prior outgroup contact. They also reported more negative affect, more unfavourable stereotypes and attributions, more contempt and less empathic concern and perspective taking than did Leavers. There was also a significant difference in education, with Remainers overall reporting higher levels of completed education, consistent with previous research[99]. There were no significant differences between groups in age or willingness to compromise.

**Tests for selective attrition.** We compared participants who at the end of the T1 survey indicated that they would like to continue with the study (i.e., to participate in the interaction; $N$ = 484) with participants who indicated that they would not like to participate further ($N$ = 102) in terms of demographic and attitudinal variables (see Supplementary Results 2). We detected selective attrition for a range of variables. Remainers were more likely to drop out than Leavers, as were older compared to younger people, and participants who reported less prior outgroup contact and lower contact quality. Those who decided to leave the study were also more strongly identified with their group, rated the outgroup as less intelligent, made more unfavourable attributions about

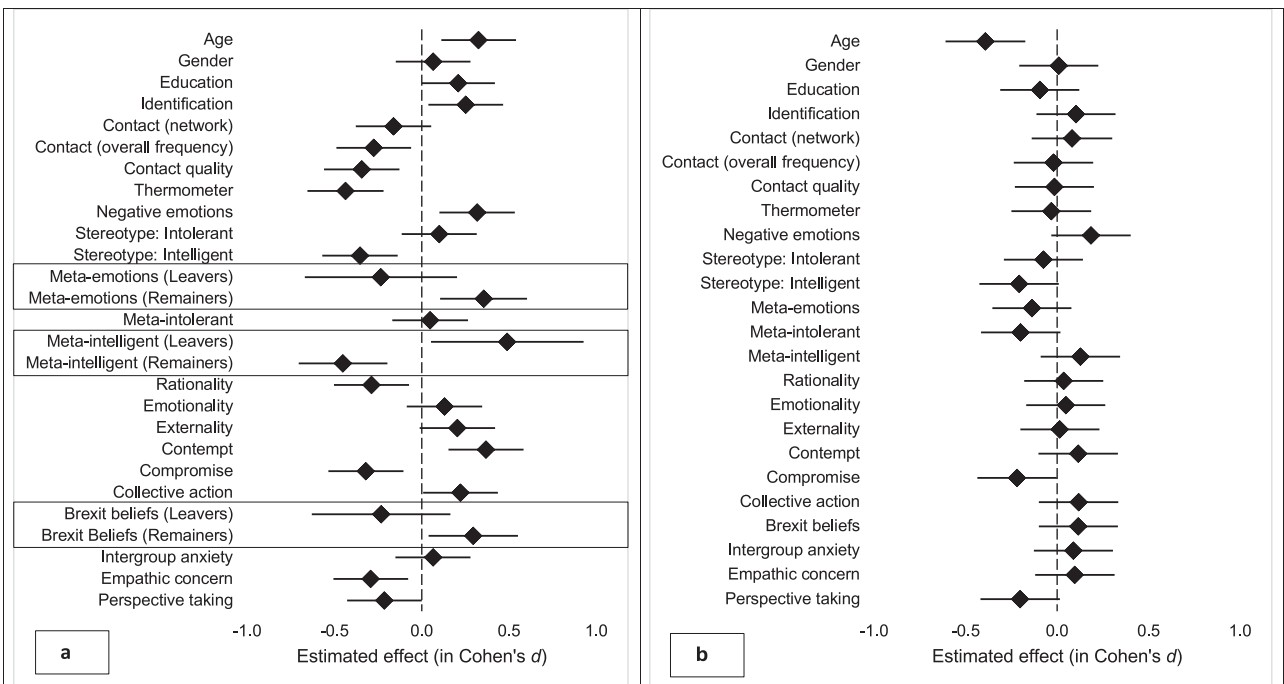

**Fig. 2 | Tests for selective attrition.** Points show the estimated effect sizes for study drop-outs. Panel 2**a** shows participants who declined to continue at T1 ($N$ = 484 participants) relative to participants who opted to continue ($N$ = 102 participants). Separate effect sizes are shown for Leavers ($N$ = 169 participants who were continuers, $N$ = 23 participants who were drop-outs) and Remainers ($N$ = 315 participants who were continuers, $N$ = 79 participants who were drop-outs) for effects that were moderated by group. Panel 2**b** shows participants who completed the interaction ($N$ = 120 participants) relative to those who failed to respond or show among those who opted to continue at T1 ($N$ = 262 participants).

the outgroups' views, reported higher levels of contempt and negative emotions, and lower levels of warmth, perspective taking, and empathic concern. Compared to continuers, drop-outs were also significantly less willing to compromise and more likely to engage in collective action on behalf of their group. There were three significant differences in predictors of drop-out between Remainers and Leavers: While Leavers who thought the outgroup viewed them as less intelligent were more likely to continue with the study, the opposite was true for Remainers. Brexit beliefs also favoured dropouts for Remainers but not Leavers. Moreover, negative meta-emotions favoured drop-out for Remainers but not for Leavers. Overall, these findings indicate that those with higher levels of attachment to their group and more negative outgroup orientations were more likely to drop out (see Fig. 2a). We also compared participants who declared at T1 that they would like to participate but failed to respond or show ($N = 262$) with participants who participated in the interaction ($N = 120$; see Fig. 2b).

The analyses indicated that younger people and those less willing to compromise with the outgroup were more likely to drop out at the later stages. There were no significant differences between Leavers and Remainers.

## Main analyses: intervention effects
Leaver-Remainer differences on study variables at T1 for the intervention sample are reported in the Supplementary Results 3.

**Comparison of intervention and control interaction ratings.** There were no significant effects of condition or group, nor were there significant interactions between group and condition for positive and negative state affect following the interaction, experienced quality of the interaction, equality-based respect during the interaction, or perceived similarity to the interaction partner. Overall, this suggests that the control and intervention conditions did not differ statistically significantly from one another in the experienced quality and that any experimental effects can therefore not be attributed to the experience of a positive interaction *per se*. Detailed results are reported in the Supplementary Results 4.

**Short-term effects.** Means and Standard deviations by condition and group for each time point are shown in the Supplementary Results 5. We tested for experimental effects on our focal outcomes in a series of three-level models, accounting for nesting of measurement points (level 1)

**Table 2 | Interaction terms for time (T1-T2) and experimental condition and for time, experimental condition and group for all outcome variables**

| Variable | Interaction terms | |
|---|---|---|
| | **Time × Condition** | **Time × Condition × Group** |
| **Outgroup affect and cognition** | | |
| Feeling thermometer | **B = 9.61, SE = 3.45, 95% CI [2.85, 16.38], Z = 2.79, p = 0.005, ΔR² = 0.01** | B = −2.90, SE = 6.22, 95% CI [−15.09, 9.29], Z = −0.47, p = 0.641, ΔR² = 0.001 |
| Negative emotions | B = −0.14, SE = 0.19, 95% CI [−0.52, 0.23], Z = −0.75, p = 0.451, ΔR² = 0.001 | B = 0.12, SE = 0.32, 95% CI [−0.50, 0.75], Z = 0.39, p = 0.699, ΔR² = 0.00 |
| Stereotype: Intolerant | B = −0.26, SE = 0.36, 95% CI [−0.95, 0.44], Z = −0.72, p = 0.472, ΔR² = 0.001 | B = 0.09, SE = 0.64, 95% CI [−1.16, 1.33], Z = 0.13, p = 0.893, ΔR² = 0.00 |
| Stereotype: Intelligent | B = 0.44, SE = 0.31, 95% CI [−0.17, 1.06], Z = 1.41, p = 0.159, ΔR² = 0.001 | B = −1.03, SE = 0.55, 95% CI [−2.12, 0.05], Z = −1.87, p = 0.062, ΔR² = 0.005 |
| Rationality | B = 0.30, SE = 0.22, 95% CI [−0.13, 0.73], Z = 1.36, p = 0.173, ΔR² = 0.002 | **B = −0.83, SE = 0.39, 95% CI [−1.60, −0.06], Z = −2.12, p = 0.034, ΔR² = 0.003** |
| Emotionality | **B = −0.65, SE = 0.31, 95% CI [−1.26, −0.04], Z = −2.08, p = 0.038, ΔR² = 0.001** | B = 0.40, SE = 0.60, 95% CI [−0.78, 1.59], Z = 0.67, p = 0.503, ΔR² = 0.001 |
| Externality | **B = −0.59, SE = 0.28, 95% CI [−1.14, −0.04], Z = −2.09, p = 0.037, ΔR² = 0.001** | B = 0.89, SE = 0.49, 95% CI [−0.08, 1.85], Z = 1.80, p = 0.072, ΔR² = 0.00 |
| Contempt | B = −0.34, SE = 0.19, 95% CI [−0.72, 0.04], Z = −1.76, p = 0.079, ΔR² = 0.003 | B = 0.38, SE = 0.39, 95% CI [−0.38, 1.14], Z = 0.98, p = 0.326, ΔR² = 0.00 |
| **Meta-beliefs** | | |
| Meta-emotions | B = −0.27, SE = 0.29, 95% CI [−0.84, 0.31], Z = −0.91, p = 0.363, ΔR² = 0.001 | B = 0.57, SE = 0.58, 95% CI [−0.57, 1.71], Z = 0.98, p = 0.329, ΔR² = 0.00 |
| Meta-Intelligence | B = 0.59, SE = 0.32, 95% CI [−0.05, 1.22], Z = 1.82, p = 0.068, ΔR² = 0.01 | B = −0.05, SE = 0.61, 95% CI [−1.25, 1.16], Z = −0.08, p = 0.940, ΔR² = 0.00 |
| Meta-Intolerance | B = −0.50, SE = 0.41, 95% CI [−1.30, 0.31], Z = −1.21, p = 0.228, ΔR² = 0.002 | B = −0.46, SE = 0.73, 95% CI [−1.88, 0.97], Z = −0.63, p = 0.528, ΔR² = 0.00 |
| **Political outcomes** | | |
| Compromise | **B = 0.35, SE = 0.18, 95% CI [0.004, 0.69], Z = 1.98, p = 0.047, ΔR² = 0.003** | B = 0.01, SE = 0.42, 95% CI [−0.82, 0.83], Z = 0.02, p = 0.985, ΔR² = 0.00 |
| Brexit beliefs | B = −0.14, SE = 0.14, 95% CI [−0.41, 0.14], Z = −0.98, p = 0.329, ΔR² = 0.001 | B = 0.46, SE = 0.28, 95% CI [−0.09, 1.00], Z = 1.64, p = 0.102, ΔR² = 0.001 |
| Collective action | B = −0.09, SE = 0.31, 95% CI [−0.70, 0.51], Z = −0.30, p = 0.765, ΔR² = 0.00 | B = −0.77, SE = 0.60, 95% CI [−1.95, 0.40], Z = −1.29, p = 0.198, ΔR² = 0.001 |
| **Traditional contact mediators** | | |
| Intergroup anxiety | B = 0.02, SE = 0.27, 95% CI [−0.52, 0.55], Z = 0.07, p = 0.948, ΔR² = 0.00 | B = 0.60, SE = 0.49, 95% CI [−0.36, 1.55], Z = 1.23, p = 0.220, ΔR² = 0.001 |
| Empathy | B = 0.23, SE = 0.24, 95% CI [−0.24, 0.70], Z = 0.97, p = 0.331, ΔR² = 0.002 | B = 0.10, SE = 0.49, 95% CI [−0.86, 1.07], Z = 0.21, p = 0.835, ΔR² = 0.00 |
| Perspective taking | B = 0.19, SE = 0.28, 95% CI [−0.37, 0.75], Z = 0.67, p = 0.504, ΔR² = 0.001 | B = −0.18, SE = 0.48, 95% CI [−1.11, 0.76], Z = −0.38, p = 0.707, ΔR² = 0.00 |

Statistically significant interactions are printed in bold. $R^2$s (and their differences) were computed following[102].

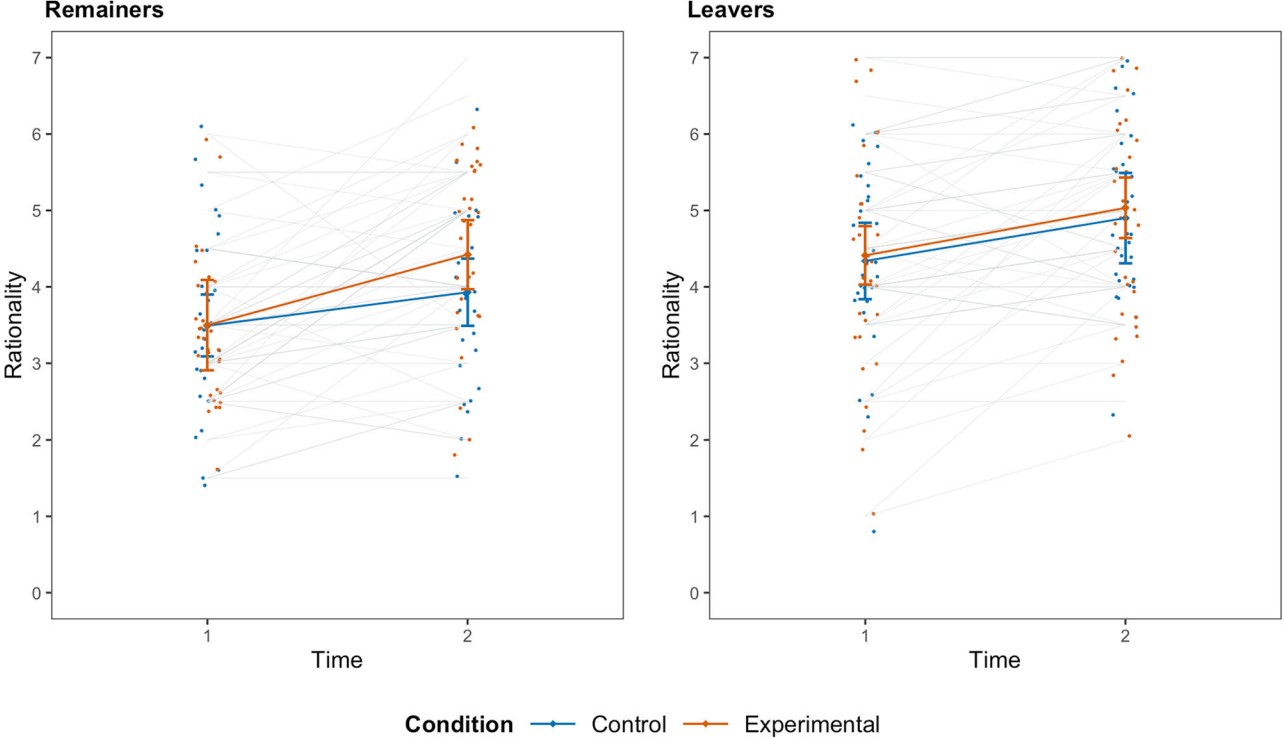

**Fig. 3 | Short-term intervention effects on rationality beliefs by group.** Raw data points, means, and 95% confidence intervals for T1 and T2 ratings of outgroup rationality for the intervention (red) and control (blue) group, separately for Remainers ($N = 60$ participants, left panel) and Leavers ($N = 60$ participants, right panel).

within individuals (level 2), and nesting of individuals within dyads (level 3). Table 2 presents the interaction terms for all outcome variables.

There were statistically significant interactions between time and condition for the feeling thermometer, emotionality, externality, and willingness to compromise. Consistent with hypotheses, and demonstrating a positive short-term effect of our intervention, participants in the treatment condition exhibited a greater increase in warmth toward the outgroup ($B = 10.23$, $SE = 2.51$, 95% $CI$ [5.30, 15.15], $Z = 4.07$, $p < 0.001$, $d = 0.63$) than participants in the control condition ($B = 1.03$, $SE = 2.80$, 95% $CI$ [−4.47, 6.52], $Z = 0.37$, $p = 0.715$, $d = 0.02$). Participants in the treatment group also reduced their attributions of emotionality ($B = −0.85$, $SE = 0.22$, 95% $CI$ [−1.28, −0.43], $Z = −3.94$, $p < 0.001$, $d = −0.52$) and externality ($B = −0.79$, $SE = 0.18$, 95% $CI$ [−1.14, −0.44], $Z = −4.43$, $p < 0.001$, $d = −0.58$) as sources of outgroup beliefs to a greater extent than participants in the control group ($B = −0.21$, $SE = 0.21$, 95% $CI$ [−0.62, 0.20], $Z = −0.99$, $p = 0.320$, $d = −0.13$, and $B = −0.20$, $SE = 0.22$, 95% $CI$ [−0.63, 0.23], $Z = −0.92$, $p = 0.355$, $d = −0.09$, respectively). Similarly, willingness to compromise with the outgroup increased significantly in the intervention ($B = 0.56$, $SE = 0.12$, 95% $CI$ [0.32, 0.80], $Z = 4.60$, $p < 0.001$, $d = 0.68$) but not in the control ($B = 0.21$, $SE = 0.13$, 95% $CI$ [−0.04, 0.46], $Z = 1.63$, $p = 0.103$, $d = 0.15$) group. There were no statistically significant interactions between time and condition for any of the other outcome variables. However, there was a statistically significant interaction between time, condition, and group for rationality attributions, indicating that the effects of the intervention varied by group. There was no statistically significant difference between the intervention ($B = 0.48$, $SE = 0.20$, 95% $CI$ [0.09, 0.86], $Z = 2.43$, $p = 0.015$, $d = 0.58$) and control ($B = 0.59$, $SE = 0.22$, 95% $CI$ [0.16, 1.03], $Z = 2.68$, $p = 0.007$, $d = 0.41$) groups for Leavers. For Remainers, changes in rationality attributions were significantly greater in the intervention ($B = 1.10$, $SE = 0.22$, 95% $CI$ [0.66, 1.53], $Z = 4.96$, $p < 0.001$, $d = 1.02$) compared to the control ($B = 0.38$, $SE = 0.21$, 95% $CI$ [−0.02, 0.78], $Z = 1.85$, $p = .064$, $d = 0.28$) group. No other 3-way interactions were statistically significant. Figure 3 visualizes the 3-way interaction for rationality attributions.

**Longer-term effects.** Table 3 present the interaction terms and simple slopes for outgroup cognition and affect, meta-beliefs, political outcomes, and traditional contact outcomes, respectively. There were no statistically significant interactions between time and condition, or between time condition and group. Thus, our hypotheses on the effects of contact on aspects of affective polarization were not supported when the longer-term outcomes (one month after the intervention) were considered.

Figure 4 displays raw data by timepoint and condition for variables that exhibited statistically significant short-term effects (feeling thermometer, emotionality and externality attributions, and willingness to compromise with the outgroup). Full model results are presented in the Supplementary Results 6.

## Discussion

Intergroup contact is viewed as one of the most promising interventions to reduce affective polarization[5,8,39] and its main tenets underpin many initiatives to bridge political divides. Experimental work aiming to establish contact's causal effects on intergroup affect and cognition in the political domain, and to determine what form contact between political opponents should take to be effective, is only beginning to emerge[38–41]. Our research makes several contributions to scholarship and practice on this issue. We developed a one-off intervention with a theoretical basis in the literature on intergroup contact[23–30] and dialogue[54–57], and the work on intergroup understanding[58–64]. We demonstrate that this intervention can be applied in an online context and evaluate its effectiveness in a randomized controlled trial with repeated measurement, using a strong control group to isolate the effects of intergroup contact from the effects of experiencing a positive interaction per se. Our design allowed us to evaluate both selective drop-out of participants as well as the causal effects of the intervention in the short- and longer-term. It thus addresses weaknesses of existing studies on the effects of contact, which lacked randomization[38], included confounding variables[39], or relied on no-treatment controls[40]. We do this in a context of affective polarization (Brexit in the UK) that has received relatively little research attention.

Our findings have implications for both future research and interventions that aim to reduce affective polarization. First, we found that

**Table 3 | Interaction terms for time (T1–T3) and experimental condition and for time, experimental condition and group for all outcome variables**

| Variable | Interaction terms | |
| --- | --- | --- |
| | **Time × Condition** | **Time × Condition × Group** |
| **Outgroup affect and cognition** | | |
| Feeling thermometer | B = −0.77, SE = 3.94, 95% CI [−8.48, 6.95], Z = −0.19, p = 0.846, ΔR² = 0.00 | B = −1.40, SE = 7.21, 95% CI [−15.53, 12.74], Z = −0.19, p = 0.846, ΔR² = 0.00 |
| Negative emotions | B = −0.13, SE = 0.14, 95% CI [−0.40, 0.15], Z = −0.92, p = 0.356, ΔR² = 0.001 | B = 0.29, SE = 0.30, 95% CI [−0.31, 0.88], Z = 0.95, p = 0.343, ΔR² = 0.001 |
| Stereotype: Intolerant | B = 0.20, SE = 0.36, 95% CI [−0.50, 0.91], Z = 0.57, p = 0.570, ΔR² = 0.001 | B = 0.97, SE = 0.65, 95% CI [−0.31, 2.25], Z = 1.48, p = 0.138, ΔR² = 0.004 |
| Stereotype: Intelligent | B = 0.06, SE = 0.32, 95% CI [−0.57, 0.69], Z = 0.20, p = 0.844, ΔR² = 0.00 | B = 0.93, SE = 0.75, 95% CI [−0.54, 2.40], Z = 1.24, p = 0.214, ΔR² = 0.01 |
| Rationality | B = 0.21, SE = 0.23, 95% CI [−0.23, 0.66], Z = 0.95, p = 0.340, ΔR² = 0.01 | B = 0.40, SE = 0.47, 95% CI [−0.54, 1.32], Z = 0.83, p = 0.405, ΔR² = 0.00 |
| Emotionality | B = 0.29, SE = 0.34, 95% CI [−0.37, 0.95], Z = 0.85, p = 0.394, ΔR² = 0.003 | B = −0.07, SE = 0.70, 95% CI [−1.43, 1.30], Z = −0.10, p = 0.924, ΔR² = 0.00 |
| Externality | B = 0.05, SE = 0.30, 95% CI [−0.54, 0.65], Z = 0.17, p = 0.865, ΔR² = 0.00 | B = −0.25, SE = 0.66, 95% CI []−1.56, 1.05], Z = −0.38, p = 0.702, ΔR² = 0.00 |
| Contempt | B = 0.09, SE = 0.20, 95% CI [−0.29, 0.48], Z = 0.48, p = 0.634, ΔR² = 0.001 | B = 0.02, SE = 0.53, 95% CI [−1.03, 0.89], Z = 0.03, p = 0.977, ΔR² = 0.00 |
| **Meta-beliefs** | | |
| Meta-emotions | B = 0.07, SE = 0.31, 95% CI [−0.54, 0.67], Z = 0.22, p = 0.823, ΔR² = 0.00 | B = −0.34, SE = 0.60, 95% CI [−1.51, 0.84], Z = −0.56, p = 0.574, ΔR² = 0.00 |
| Meta-stereotype: Intelligent | B = −0.47, SE = 0.28, 95% CI [−1, 0.06], Z = −1.70, p = 0.089, ΔR² = 0.002 | B = −0.62, SE = 0.91, 95% CI [−2.40, 1.15], Z = −0.69, p = 0.491, ΔR² = 0.002 |
| Meta-stereotype: Intolerant | B = −0.53, SE = 0.39, 95% CI [−1.30, 0.24], Z = −1.36, p = 0.175, ΔR² = 0.003 | B = 1.13, SE = 0.90, 95% CI [−0.62, 2.89], Z = 1.27, p = 0.206, ΔR² = 0.01 |
| **Political outcomes** | | |
| Compromise | B = 0.03, SE = 0.21, 95% CI [−0.38, 0.44], Z = 0.13, p = 0.894, ΔR² = 0.00 | B = −0.43, SE = 0.38, 95% CI [−1.18, 0.32], Z = −1.11, p = 0.266, ΔR² = 0.003 |
| Brexit beliefs | B = −0.05, SE = 0.14, 95% CI [−0.32, 0.22], Z = −0.37, p = 0.710, ΔR² = 0.00 | B = −0.01, SE = 0.31, 95% CI [−0.61, 0.59], Z = −0.03, p = 0.974, ΔR² = 0.00 |
| Collective action | B = 0.10, SE = 0.38, 95% CI [−0.64, 0.84], Z = 0.26, p = 0.792, ΔR² = 0.00 | B = 0.21, SE = 0.53, 95% CI [−0.83, 1.26], Z = 0.40, p = 0.692, ΔR² = 0.002 |
| **Traditional contact mediators** | | |
| Intergroup anxiety | B = −0.13, SE = 0.26, 95% CI [−0.64, 0.37], Z = −0.52, p = 0.603, ΔR² = 0.00 | B = 0.20, SE = 0.55, 95% CI [−0.87, 1.27], Z = 0.37, p = 0.713, ΔR² = 0.02 |
| Empathy | B = 0.11, SE = 0.28, 95% CI [−0.44, 0.65], Z = 0.38, p = 0.705, ΔR² = 0.00 | B = −0.15, SE = 0.52, 95% CI [−1.16, 0.85], Z = −0.29, p = 0.773, ΔR² = 0.00 |
| Perspective taking | B = 0.11, SE = 0.26, 95% CI [−0.40, 0.61], Z = 0.41, p = 0.685, ΔR² = 0.001 | B = −0.19, SE = 0.56, 95% CI [−1.29, 0.90], Z = −0.34, p = 0.731, ΔR² = 0.002 |

drop-out from our study was systematic, such that participants who were more strongly committed to their group and have more negative orientations toward the political outgroup were less willing to participate in the interaction. On the one hand, this might undermine the wider generalizability of our findings. Given that those who are ideologically most extreme tend to benefit the most from contact with outgroup members[89], it is possible that the current study underestimates the effects of contact. On the other hand, estimating treatment effects among those who voluntarily choose to participate provides more realistic effect sizes for applications in the "real world", as those most strongly invested in their group memberships are less likely to conform to requests to engage respectfully with the outgroup[100].

This has important practical implications as it suggests that it will be challenging to involve those who are at the ideological extremes in interventions to reduce intergroup animus. There is theoretical merit in examining how this segment of the population can be motivated to participate, and how they respond to such interventions. Our finding that intergroup cognition and affect shape actual behaviour such as avoidance of intergroup contact underlines the relevance of studying factors that impact on affective polarization. From a practical perspective, bringing together those segments of the population who have more moderate views is important as this has greater prospect to find common ground and disconfirm negative outgroup views. Giving moderate viewpoints a platform for respectful exchange might reduce the undue influence of the most extreme voices on perceived social norms and thereby strengthen the centre of society. Nonetheless, our findings suggest that contact interventions might be difficult to bring about in contexts of extreme polarization.

Second, we observed significant positive short-term effects of our intervention on only a subset of outcome variables and found no evidence for its longer-term effects. Compared to the control group, our intervention sample exhibited a greater short-term increase in warmth toward the outgroup, reduced attributions of emotionality and externality as the sources of outgroup positions and increased willingness to compromise with the outgroup over political issues. These findings are in line with our hypotheses and the theoretical contact literature more generally and provide some support that contact with political opponents that involves respectful dialogue can increase intergroup warmth, alleviate unfavourable attributions of why outgroup members take a particular position, and increase openness to compromise, at least in the short term. The latter finding is particularly

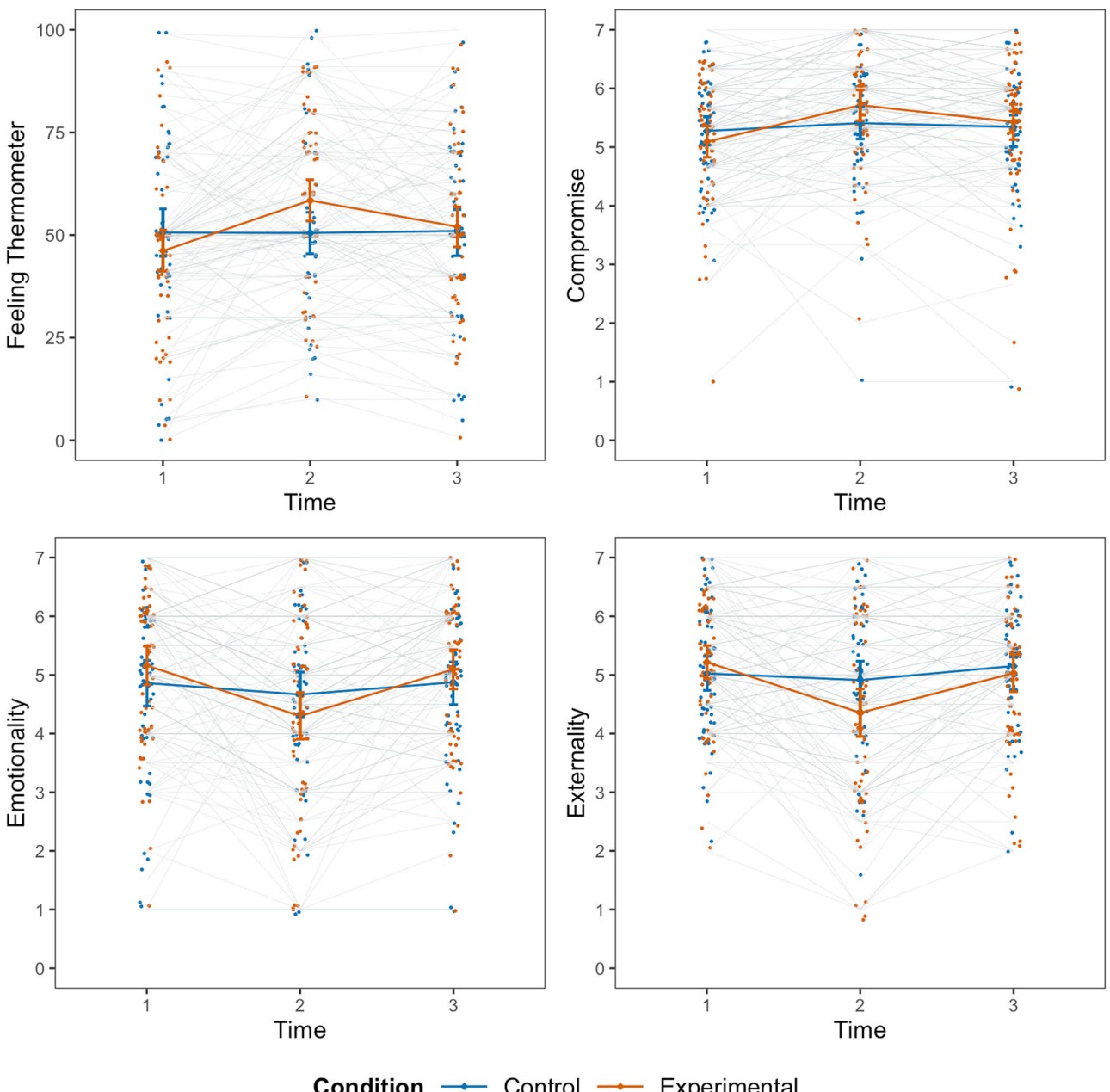

**Fig. 4 | Short- and longer-term intervention effects on feeling thermometer, compromise willingness, emotionality and externality ratings.** Raw data, means, and 95% confidence intervals for each time point for the intervention ($N = 60$ participants, red) and control ($N = 60$ participants, blue) group.

noteworthy as it suggests that contact interventions that involve dialogue could have positive 'downstream effects' on commitment to democratic principles. There was also one statistically significant interaction with group (Leaver vs. Remainer) for rationality attributions which is worth highlighting. This indicated that the intervention had a stronger effect for Remainers than for Leavers. This is consistent with the literature suggesting that those with more negative views to begin with change their views to a greater extent following contact[88,89] and highlights the importance of considering group membership when investigating contact effects[51].

**Limitations**
Overall, evidence for the success of our intervention can at best be considered mixed, as there were no statistically significant intervention effects for any of the other outcome variables, nor was there any evidence that these effects were sustained over the course of one month. This severely limits the

direct applicability of our intervention and calls for more systematic research. It is possible that one-off interventions such as the current one might not be sufficient to engender enduring change. Future research therefore needs to design and evaluate interventions that enable sustained dialogue over longer periods of time. Given the practical effort involved in setting up interactions such as these, and the limited direct societal impact of contact interventions, future research should also explore the possibility to automatize the moderation of interactions (see ref. 38), which was done by a trained moderator in our study, to increase scalability. Future research might also explore the effects of sustained vicarious contact, for example by exposing people to respectful cross-party dialogue through media. This might have impacts far beyond its direct participants by shaping social norms that support intergroup dialogue[101]. Dialogue interventions such as the one tested here could also be combined with other initiatives such as sustained media messaging that encourages a pro-dialogue social norm.

## Conclusions

Affective polarization is on the rise in around the world[5–7]. It divides societies and weakens democracy[8–10]. This trend underlines the importance of creating safe spaces that foster the openness necessary to engage with complex social problems that require a plurality of voices. Our research contributes new insights into the question of whether interventions involving structured contact between political adversaries under conditions enabling respectful mutual exchange can reduce affective polarization and promote willingness to cooperate compromise over political issues. While our intervention produced several significant positive short-term effects on intergroup affect, cognition, and compromise willingness, we did not find any significant long-term effects on outcome variables. Moreover, our research points to important challenges in bringing about face-to-face interactions between political opponents, particularly those with more extreme outgroup views. These mixed findings suggest that theory-based contact interventions such as the current one require further investigation before they can be more widely adopted. Our work provides a framework for future research, which could focus in on different elements of our intervention and explore combining contact with norm-based interventions to produce longer-term effects.

## Data availability

Data and study materials have been deposited in Open Science Framework (https://osf.io/ra6yj/).

## Code availability

The analyses codes are available through the Open Science Framework (https://osf.io/ra6yj/).

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

## Acknowledgements

This research was funded by the British Academy/Leverhulme Small Research Grant "Bridging the divide: Piloting an intergroup contact intervention in the context of Brexit" (SRG19\191421), awarded to the first (Principal Investigator) and second author. The funders had no role in study design, data collection and analysis, decision to publish or preparation of the manuscript. We thank Harris Siderfin for assisting us in the recruitment process.

## Author contributions

N.T. conceived of and led the project, including acquisition of funding, overseeing the data collection, ethical approval process and data curation, and wrote the manuscript. N.T. and M.D.B. contributed to the design of the study, ethics application, and interpretation of data and writing. P.G. and N.T. analysed the data. S.B. and C.R. conducted the interactions and collected the data. All authors revised and approved the manuscript.

## Competing interests

The authors declare no competing interests.
