## [Peer Review file · Communications Psychology]

A post-Brexit intergroup contact intervention reduces affective polarization between Leavers and Remainers short-term

Corresponding Author: Dr Nicole Tausch

Version 0:

Decision Letter:

Dear Dr Tausch,

Thank you for your patience during the peer-review process. Your manuscript titled "The Causal Effect of an Intergroup Contact Intervention on Affective Polarization around Brexit: A Randomized Controlled Trial" has now been seen by 3 reviewers, and I include their comments at the end of this message. They find your work of interest but raised some important points. We are interested in the possibility of publishing your study in *Communications Psychology*, but would like to consider your responses to these concerns and assess a revised manuscript before we make a final decision on publication.

We therefore invite you to revise and resubmit your manuscript, along with a point-by-point response to the reviewers. Please highlight all changes in the manuscript text file.

Editorially, we ask you to prioritize the following issues in your revision:

- 1) The reviewers (esp #1 and #2) express concerns about the sample / power. Please provide a sensitivity analysis that reports the sample's power to detect the smallest effect size of interest (not the reported effect size); please also comply with the request for an additional attrition analysis.
- 2) The journal upholds clear standards for statistics reporting and interpretation, which are aligned with some of the reviewer comments. Differences between statistical significance between different tests may not be interpreted in the absence of a statistically significant interaction effect. Any result at $P > 0.05$ may not be interpreted, unless the alpha-level was a priori set to $\alpha = 0.1$ with an appropriate justification (see also concerns about the transparency about the nature of the preregistration). Non-significant results may not be interpreted on the basis of NHST as evidence for the absence of an effect. Instead please report Bayes Factors or equivalence tests.
- 3) Please address the reviewers' concerns regarding the nature of the control analysis and incorporate potential caveats in the limitations section in the Abstract. Similarly, please ensure that the results are presented close to the data, i.e., avoiding overinterpretation or claims of generalization throughout.

I am attaching an Editorial Requests Table that details critical reporting requirements for the revised manuscript. Please attend to each item and ensure your manuscript is fully compliant. We are requesting that your manuscript aligns with these requirements as this facilitates the evaluation of your manuscript, reducing delays in re-review and potential future acceptance. If your revised manuscript is not aligned with these requests on major issues, such as those concerning statistics, it may be returned to you for further revisions without re-review. Additional information can be found in our style and formatting guide Communications Psychology formatting guide.

Please use the following link to submit your

- revised manuscript,
- point-by-point response to the referees' comments,
- cover letter (as a separate document),
- the Editorial Policy Checklist (see below),
- the Reporting Summary (see below), and
- the completed Editorial Request Table (attached):

Link Redacted

Best regards,

Hannah Nam

Hannah Nam, PhD
Editorial Board Member
Communications Psychology
orcid.org/0000-0003-3006-0584

REVIEWER REPORTS:

Reviewer #1 (Remarks to the Author):

Review of MS COMMSPSYCHOL-24-0083-T

by Tausch et al.

Reviewer: Stephan Lewandowsky

Summary and Overall Recommendation

The paper reports a study that explored an intervention during face-to-face online conversations to reduce the affective polarization around the Brexit divide (Leavers vs. Remainers) in the UK. The intervention consisted of multiple stages, each of which was informed by existing literature on how to facilitate inter-group contact, introduce respect, perspective-giving, and so on. A variety of relevant measures were taken before the intervention (baseline); immediately after the intervention, and about one month later. In a control group, participants experienced the same intervention but did not discuss Brexit (but the exit of Prince Harry and Meghan Markle from the British royal family). The authors report several beneficial effects of their intervention immediately after the experiment, although most of those were not sustained over time. The authors also examine the predictors of attrition (i.e., participants who chose not to proceed beyond the pretest), finding that dropouts were more strongly committed to their group and had more negative orientations toward the outgroup.

Turning to evaluation, there is much to like about the paper. The review of the theoretical background is thorough and informative, the intervention is carefully designed and embedded in prior theoretical work, and the data analysis is thorough and informative. I am therefore positively inclined towards this manuscript, although there are a few issues that prevent me from endorsing publication at the moment. I expand on those below and I expect that they ought to be resolvable during a revision.

Major points

1. The authors understandably make much of the Brexit divide in the UK. That divide undoubtedly exists but there are some recent data suggesting that the conventional partisan divide (Labour vs. Conservative) is considerably stronger (Prike et al; DOI 10.1098/rsos.220508). It is therefore unfortunate that the current study apparently did not seek participants' party affiliation or preferences.

2. I was concerned by the number of different measures in the study. This provides an opportunity for selective reporting and/or selective focus on the measures that "work". I checked the preregistration and I was unable to find a highly specific analysis plan that might guard against the problem – the original funding proposal was fairly general. I encourage the authors to address this potential criticism, perhaps in the response letter.

3. The final sample size (N=120, so only ~60 per group) was relatively small, given the large pool of participants (N=484) available for the intervention. Why? There are obvious risks with running under-powered studies and this issue should be addressed.

Detailed comments
[page#]:[para#]:[line#]

3:3:4 "has allows"  "allows".

6:2:1 "ases"  "assess".

14:Fig 3 There should be error bars on the points and the font size in caption and axes labels should be increased.

21:2:5 "contract"  "contact".

26:4:2 "emotivational" ?

Reviewer #2 (Remarks to the Author):

This is a well-written manuscript that makes a clear case for the need for more research, especially outside the US, on the effect of intergroup contact in the political domain on reducing affective polarization. The study uses a rigorous design with careful measurement to produce causal estimates of the impact of a well-informed intervention. However, the framing of the paper and discussion of the findings, in my view, oversell the results. First, the discussion claims that there are significant effects on more outcomes than appear to be the case given the results in the tables. But more importantly, the immediate effects of the intervention all decay by one month. I would suggest the authors update the tone and reframe the discussion to be more grounded in these mixed findings and less optimistic about what they imply for our understanding of the power of intergroup contact in this domain. I also have concerns about the power of the experiment and what we can conclude in terms of generalizability which I discuss more below.

First, the discussion section states that the intervention had significant effects on stereotypes, meta-beliefs and contempt. According to Tables 2 and 3, there is no effect on stereotypes or meta-beliefs and the effect on contempt (and on meta-intelligence) is only statistically significant at the 90% threshold. I am not sure on what basis the first claim is made, and I would want the authors to qualify the claims for which the statistical significance is weaker. The authors also state that the meta-stereotypes show a statistically significant reduction in the treatment group, but again, this is only statistically significant at 90%.

Second, the authors are otherwise transparent about the fact that there are no longer-term effects of the intervention. However, they still say the results are "highly encouraging" and that the results are of considerable importance for efforts to reduce affective polarization. Given the relative intensity of the intervention and the difficulty of recruiting participants, this very ephemeral impact does not, to me, clearly imply that governments or organizations should be investing in more of this work. I wanted the authors to justify their enthusiasm about these results in light of these challenges and temper the tone of the paper in describing just how impactful these findings are and what exactly they would imply for future policy.

I was struck but the relatively small sample size of the study. In a back of the envelope calculation, the authors are decently powered to detect effect sizes of about one-half of a standard deviation. Indeed, this is about the size of the effects on the outcomes they find are statistically significant. However, that is a relatively high bar for social science experiments. Typically, we see effect sizes of between 0.2 to 0.3 standard deviations. It could be then, that some of the estimated null effects are type II errors and due to the study being underpowered. I do not think this is likely the case for the effects estimated at T3 given that many are very close to 0; however, this could be the case for some of the other outcomes estimated at T2 where effects were found to be insignificant. This matters, in part, because of the promise of learning about not just whether but how intergroup contact works. One strength of the study is the thoughtful design of the intervention and outcome measures. This should help us understand whether this very particular type of moderated discussion about political issues works, and on which outcomes. But because some of the outcomes may exhibit smaller effects, we may not know with any confidence whether the pattern of effects working on some outcomes but not others should be taken as instructive for our understanding of the mechanics of this intervention.

Related to power, it was unclear from the paper why there were only 120 participants who participated in the intervention when 484 indicated that they would like to have continued with the study. From the manuscript, it was unclear how the 484 participants got whittled down to 120 and why. After reading through the supplementary materials, I can see that the reasons include non-response or no-shows (with a minority of people not being able to be matched due to scheduling). The first two categories of reasons are especially important for thinking about generalizability. The authors nicely conduct tests for selective attrition across those who report willingness to continue in the study or not. But I also would have wanted to see this done for people who completed at least the intervention and T2 survey relative to those who dropped out.

Finally, I liked the fact that the authors had a control condition that allows us to identify a very specific aspect of intergroup contact we care about: political interactions. However, it was unclear to me just how apolitical the control condition was. Were respondents in that condition aware of the fact that they were paired with an outgroup member or someone who had a different perspective about Brexit? Or were they aware that the goal of the study was to interrogate intergroup contact along

this dimension? That would be important to know since I would expect that the findings would underestimate effects of contact if participants had some information about the intergroup nature of the exchange.

Reviewer #3 (Remarks to the Author):

I am excited about the present research for several reasons. Even though there must, by now, be thousands of studies on the relationship between intergroup contact and prejudice, Paluck et al. (2018, 2020) only found 28 studies that fulfilled the strict criteria of a randomized controlled trial and of which only 8 studies measured outcomes at least one day after the contact-based intervention. As the present research does both of those things, it addresses the urgent need for evidence concerning the causal effects of intergroup contact. Further, the intervention stands out in its adherence to best practices informed by intergroup contact theory and other theoretical perspectives, in particular, its implementation of the sequential model of intergroup contact (Pettigrew, 1998). Another strength of the present research that is perhaps less obvious to researchers outside of the field of intergroup contact research is its control condition. Similar studies tend to rely on either a no-contact control group in which participants do not interact with anyone or an ingroup-contact control group in which participants interact with an ingroup member. The former is problematic as it does not allow isolating the effect of intergroup interaction from that of any social interaction while the latter is problematic as ingroup contact itself might have an effect on relevant outcomes. Tausch et al.'s choice of a control group is elegant as participants interact with an outgroup member in both condition but are only aware of the interaction partner's group membership in the experimental condition. For those reasons, I recommend the publication of this manuscript in *Communications Psychology*.

Still, I want to offer a few comments and recommendations.

First, I think that the authors overstate the support for the contact hypothesis in the literature at this moment. For example, I was surprised that the authors support a claim about the causal effects of intergroup contact—"intergroup contact robustly reduces negative intergroup attitudes"—with evidence from correlational studies (van Assche et al., 2023). In a sense, this framing undermines the importance of the present research which stands out as one of the rare experimental studies in this field. In any case, I think it is worth noting that recent research highlighted the limited experimental evidence (for a review, see Paluck et al. 2018, 2020) and contradicting longitudinal evidence (e.g., Friehs, Bracegirdle, et al., 2024, *Soc. Psychol. Pers. Sci.*; Hodson & Meleady, 2024, *Am. Psychol.*; Sengupta et al., 2023, *Am. Psychol.*; but see Gorska & Tausch, 2023, *Soc. Psychol. Pers. Sci.*) for the causal effects of intergroup contact.

Second, the authors do not disclose that they registered the preregistration after they had collected the data (as stated in the preregistration). Readers might have different opinions on how useful preregistration (postregistration?) after data collection is so this fact should be disclosed in the main text. I, personally, do not see preregistration as particularly informative in most such cases.

Third, the authors summarize the results from Table 2 as showing that "there were significant changes from T1 to T2 on most outcome variables in the expected directions in the intervention condition, and very few effects for the control group". To me, this reads as somewhat misleading as most Time x Condition interactions are not significant. After all, the difference between "significant" and "not significant" is not itself statistically significant (<https://stat.columbia.edu/~gelman/research/published/signif4.pdf>).

Lastly, I have a few clarification questions. Do you know whether Remainers and Leavers differed in their opinions on the control topic? In a sense, this is not an unpolitical topic (as it touches on participants' attitudes concerning racism and the monarchy) and I would expect Leavers and Remainers to differ in their opinions on it. How were the participants selected from all those who agreed to participate in the interaction? How did you determine the sample size for this research?

I hope the authors will find my comments useful for improving the present manuscript. I wish them all the best for their current and future research.

Nils Reimer

EDITORIAL POLICIES

We ask that you ensure your manuscript complies with our editorial policies and reporting requirements.

To that end, we require revised manuscripts to be accompanied by two completed items: a reporting summary that collects information on study design and procedure, and an editorial policy checklist that verifies compliance with all required editorial policies.

- <https://www.nature.com/documents/nr-reporting-summary.zip>>Nature Research Reporting Summary

- <https://www.nature.com/documents/nr-editorial-policy-checklist.pdf>>Editorial Policy Checklist

All points on the policy checklist must be addressed. Your revised manuscript can only be sent back to the referees if these checklists are completed and uploaded with the revision.

Notes: If you have submitted a Stage 1 Registered Report, Review, Primer, Comment, or Perspective you do not need to submit these forms. If you have already submitted these forms, you may disregard this request.

** Visit Nature Research's author and referees' website at <http://www.nature.com/authors>>www.nature.com/authors for information about policies, services and author benefits**

If you experience problems in linking your ORCID, please contact the <http://platformsupport.nature.com/>>Platform Support Helpdesk.

Author Rebuttal letter: The author's response to these comments can be found at the end of this file.

Version 1:

Decision Letter:

Dear Dr Tausch,

Your manuscript titled "The Causal Effect of an Intergroup Contact Intervention on Affective Polarization around Brexit: A Randomized Controlled Trial" has now been seen by our reviewers, whose comments appear below. In light of their advice I am delighted to say that we are happy, in principle, to publish a suitably revised version in Communications Psychology.

We therefore invite you to revise your paper one last time to address the remaining concerns of our reviewers and a list of editorial requests. At the same time we ask that you edit your manuscript to comply with our format requirements and to maximise the accessibility and therefore the impact of your work.

EDITORIAL REQUESTS:

SUBMISSION INFORMATION:

In order to accept your paper, we require the files listed at the end of the Editorial Requests Table; the list of required files is also available at <https://www.nature.com/documents/commsj-file-checklist.pdf> .

OPEN ACCESS:

Communications Psychology is a fully open access journal. Articles are made freely accessible on publication. For further information about article processing charges, open access funding, and advice and support from Nature Research, please visit <https://www.nature.com/commpsychol/open-access>

* DATA AVAILABILITY:

Link Redacted

Best regards,

Marika

Marika Schiffer, PhD
Chief Editor
Communications Psychology

REVIEWERS' COMMENTS:

Reviewer #1 (Remarks to the Author):

Review of MS COMMSPSYCHOL-24-0083A

by Tausch et al.

Reviewer: Stephan Lewandowsky

Summary and Overall Recommendation

I reviewed this manuscript previously and my impressions at the time were already largely positive. I did, however, raise 3 major issues that prevented me from endorsing publication previously:

1. The authors seemingly exaggerated the Brexit divide in the UK relative to the partisan divide. In response, the authors correctly note that while the partisan divide had become more relevant (again) recently and is now dwarfing the Brexit divide, at the time that their data were collected the Brexit issue was still highly divisive. I accept the argument and the authors now explicitly state when the data were collected.

2. I was concerned by the number of different measures in the study which, given the absence of a strict preregistered analysis plan, might support inadvertent cherry-picking. The authors' response notes that they reported all results for all measures, including those that were nonsignificant, and that they no longer consider the "preregistration" to be a conventional preregistration. I accept the arguments.

3. The final sample size (N=120, so only ~60 per group) was relatively small, and I was concerned about the power of the study. In response, the authors conducted a thorough examination of the power issue and I believe that they have addressed this concern satisfactorily.

I have no further comments and I am happy to endorse publication of the paper.

Reviewer #2 (Remarks to the Author):

I am very satisfied with the authors' careful and comprehensive response to the initial set of reviews, both mine and from other reviewers. I support publication of this manuscript.

Reviewer #3 (Remarks to the Author):

I thank the author for engaging with my questions, comments, and suggestions. All my outstanding concerns have been addressed in the revision.

Author Rebuttal letter: The author's response to these comments can be found at the end of this file.

Dr Nicole Tausch
School of Psychology & Neuroscience
University of St Andrews
St Andrews, KY16 9JP
United Kingdom
Website
Google Scholar

Revision of “The Causal Effect of an Intergroup Contact Intervention on Affective Polarization around Brexit: A Randomized Controlled Trial” (MS COMMSPSYCHOL-24-0083-T)

Dear Reviewers,

We would like to thank you for the valuable feedback for improving our manuscript entitled “The Causal Effect of Intergroup Contact on Affective Polarization around Brexit: A Randomized Controlled Trial” for consideration of publication in *Communications Psychology*. We have now addressed all the issues you raised and revised the manuscript accordingly. Your comments and suggestions were extremely helpful in improving our paper and we hope that you will consider the revised manuscript worthy of publication. We give a detailed account of how we dealt with each point below. Of course, we are open to respond with further clarifications or to incorporate additional revisions if necessary.

Reviewer #1

The paper reports a study that explored an intervention during face-to-face online conversations to reduce the affective polarization around the Brexit divide (Leavers vs. Remainers) in the UK. The intervention consisted of multiple stages, each of which was informed by existing literature on how to facilitate inter-group contact, introduce respect, perspective-giving, and so on. A variety of relevant measures were taken before the intervention (baseline); immediately after the intervention, and about one month later. In a control group, participants experienced the same intervention but did not discuss Brexit (but the exit of Prince Harry and Meghan Markle from the British royal family). The authors report several beneficial effects of their intervention immediately after the experiment, although most of those were not sustained over time. The authors also examine the predictors of attrition (i.e., participants who chose not to proceed beyond the pretest), finding that dropouts were more strongly committed to their group and had more negative orientations toward the outgroup. Turning to evaluation, there is much to like about the paper. The review of the theoretical background is thorough and informative, the intervention is carefully designed and embedded in prior theoretical work, and the data analysis is thorough and informative. I am therefore positively inclined towards this manuscript, although there are a few issues that prevent me from endorsing publication at the moment. I expand on those below and I expect that they ought to be resolvable during a revision.

Dr Nicole Tausch
School of Psychology & Neuroscience
University of St Andrews
St Andrews, KY16 9JP
United Kingdom
Website
Google Scholar

Thank you very much for these encouraging words!

- 1. The authors understandably make much of the Brexit divide in the UK. That divide undoubtedly exists but there are some recent data suggesting that the conventional partisan divide (Labour vs. Conservative) is considerably stronger (Prike et al; DOI 10.1098/rsos.220508). It is therefore unfortunate that the current study apparently did not seek participants' party affiliation or preferences.**

Thank you for highlighting this issue; it certainly deserves attention. We agree that there is polarization both in terms political party support and Brexit position in the United Kingdom. Evidence points to the differential importance of these identities as a function of time since the 2016 referendum, the substantive issue of concern (e.g., opinions on immigration), and what exactly is being measured. For example, using a range of data sets collected between 2016 and 2019, Hobolt et al. (2021) found that Brexit identities produced affective polarization (measured in terms of stereotypes, evaluative biases etc.) that was of equal or larger intensity than that of partisanship, and that the effects of the Brexit divide cut across traditional partisanship, highlighting the importance of this divide as a source of polarization over and above partisanship. Survey data collected during the 2019 general election campaigns also showed that identification with a Brexit side was stronger than identification with a political party, with both identities having gained in strength compared to 2018 (although this has been even more so for Brexit identities; The Policy Institute King's College London, 2019). When considering polarization around the issue of immigration in 2019, the divide on restricting immigration (a key factor for voting in favour of Brexit) was larger between Remainers and Leavers (66% disagree vs. 90% agree) than between supporters of Labour vs the Conservatives (51% disagree vs. 81% agree). The British Social Attitudes survey found no evidence that political polarization between Leavers and Remainers was decreasing in 2020/21 (see also Butler, 2021). Moreover, providing evidence for continuing polarization, 2020 saw a reversal of typical patterns of political trust, with Leavers more likely than Remainers to trust the government (NatCen, 2020). As you correctly point out, however, other evidence highlights that partisan identities can produce greater divisions than Brexit identities (Pew Research Center, 2019; Prike et al., 2023). Nonetheless, the key point is that the political identities that were the focus of our research were highly relevant and produced polarization in the UK population at the time that our research was conducted (between November 2020 until April 2021), something that was also evident in our data.

Dr Nicole Tausch
School of Psychology & Neuroscience
University of St Andrews
St Andrews, KY16 9JP
United Kingdom
Website
Google Scholar

While Brexit remains a relevant issue in 2024 (National Centre for Social Research, 2024), more recently (from about 2022) there has indeed been a greater divide in terms of partisan identities and a decline in the importance of Brexit identities. Survey data suggests that a majority of Britons feel that the UK was wrong to leave the EU (56% to 32%), and that a sizeable proportion of Leave voters are regretting their vote (YouGov, 2022; The Policy Institute King's College London, 2022). Moreover, several recent scandals such as the violations of Covid restrictions (“partygate”) revealed in the Sue Gray report published in May 2022, as well as the economic shock following Liz Truss’ “mini-budget” in 2022, has brought partisan identities to the fore (e.g., British Politics at Queens, 2024).

To address any potential doubt about the suitability of the Brexit debate to study contact effects in a context of political polarization, we now highlight the timing of our study, and refer to the importance of this divide during this time, in the main manuscript on page 8:

“Data were collected between November 2020 and April 2021, a period during which Brexit continued to be a salient issue in public discourse and Brexit identities remained highly important among the British population [Butler 2021]”.

We have also included a section on this issue in the supplementary materials (Section A.3.6 Timing of study and relevance of Brexit identities).

Regarding your question about relevant measures of partisanship in our study: We did in fact collect political party support in our study by including 3 questions on past voting behaviour and future voting intentions in our T3 survey. In accordance with prior survey results, which indicated that partisanship/political orientation and Brexit identities are correlated (e.g., Conservatives (60%) and people on the political right (48%) were more likely to vote Leave. Liberal Democrats (70%), Labour (53%) and people on the political left (66%) were more likely to vote Remain; Pew Research Center, 2019), our data show an alignment between partisanship and Brexit position. Leavers reported more frequently voting for the Conservatives, Remainers reported more frequently voting for Labour or the Liberal Democrats (see Table below).

Past voting and voting intentions in the UK general election, by group

		Remain	Leave
2015 General Election	Conservatives	28.3	68.3
	Labour	46.7	13.3
	Liberal Democrats	15	0
2019 General Election	Conservatives	20	75

Dr Nicole Tausch
School of Psychology & Neuroscience
University of St Andrews
St Andrews, KY16 9JP
United Kingdom
Website
Google Scholar

	Labour	46.7	10
	Liberal Democrats	13.3	1.7
Next General Election	Conservatives	20	61.7
	Labour	43.3	13.3
	Liberal Democrats	10	1.7

Items: *Which political party did you vote for in the 2015 / 2019 general election? Which political party would you vote for in the next general election?*

We have added this information to the section detailing sample characteristics in the Supplement (Section A1) for interested readers.

References

British Politics at Queens (2024). One PM too many? Brexit, Party gate, and the fall of the Conservative Government. <https://blogs.qub.ac.uk/pb-happ/2024/03/06/one-pm-too-many-brexit-party-gate-and-the-fall-of-the-conservative-government/>

Butler, P. (2021). British leavers and remainers as polarised as ever, survey finds. <https://www.theguardian.com/politics/2021/oct/21/british-leavers-and-remainers-as-polarised-as-ever-survey-finds>

Prike, T., Reason, R., Ecker, U. K., Swire-Thompson, B., & Lewandowsky, S. (2023). Would I lie to you? Party affiliation is more important than Brexit in processing political misinformation. Royal Society Open Science, 10(2), 220508. <https://doi.org/10.1098/rsos.220508>

Pew Research Center (2019). Brexit divides the UK, but partisanship and ideology are still key factors <https://www.pewresearch.org/short-reads/2019/10/28/brexit-divides-the-uk-but-partisanship-and-ideology-are-still-key-factors/>

National Centre for Social Research, (2020). The British Social Attitudes survey <https://lgiu.org/briefing/natcen-british-social-attitudes-survey-2020/>

National Centre for Social Research (2024) Identity issues now a key dividing line in Britain's politics. <https://natcen.ac.uk/news/identity-issues-now-key-dividing-line-britains-politics>

The Policy Institute King's College London (2019). How polarised is the 2019 general election?

Dr Nicole Tausch
School of Psychology & Neuroscience
University of St Andrews
St Andrews, KY16 9JP
United Kingdom
Website
Google Scholar

<https://www.kcl.ac.uk/policy-institute/assets/how-polarised-is-the-2019-general-election.pdf>

The Policy Institute King's College London (2022). Leavers 'more likely to regret Brexit vote' than Remainers, study finds. <https://www.kcl.ac.uk/news/leavers-more-likely-to-regret-brexit-vote-than-remainers-study-finds>

The Week Staff. (February 16, 2023 Thursday). Brexit: what changed after the UK pulled out of the EU. The Week.

<https://advance.lexis.com/api/document?collection=news&id=urn:contentItem:67JT-R5X1-JB34-4239-00000-00&context=1519360>.

YouGov (2022). One in five who voted for Brexit now think it was the wrong decision.

https://yougov.co.uk/politics/articles/44445-one-five-who-voted-brexit-now-think-it-was-wrong-d?redirect_from=%2Ftopics%2Fpolitics%2Farticles-reports%2F2022%2F11%2F17%2Fone-five-who-voted-brexit-now-think-it-was-wrong-d

- 2. I was concerned by the number of different measures in the study. This provides an opportunity for selective reporting and/or selective focus on the measures that “work”. I checked the preregistration and I was unable to find a highly specific analysis plan that might guard against the problem – the original funding proposal was fairly general. I encourage the authors to address this potential criticism, perhaps in the response letter.**

We aimed to assess different aspects of affective polarization and our choice of outcome measures was informed by both the social-psychological literature on intergroup relations and intergroup contact as well as the research on political polarization. It was important to us to go beyond assessing general affective orientations toward the outgroup (e.g., feeling thermometer) and include measures of variables that might give insights into cognitive and affective processes associated with polarization (e.g., rationality attributions, contempt, perspective taking, meta-beliefs), as well as capture potentially politically relevant outcome measures such as willingness to compromise and to take collective action. The latter goal was informed by findings in the contact literature suggesting that improved attitudes do not necessarily translate into consequential outcomes such as sharing resources or policy support, and that contact can undermine ingroup-supporting action tendencies (e.g., Saguy, Tausch, Dovidio, & Pratto, 2009). We would like to clarify here that we report the results on all outcome measures that were measured identically at

Dr Nicole Tausch
School of Psychology & Neuroscience
University of St Andrews
St Andrews, KY16 9JP
United Kingdom
Website
Google Scholar

all three time points. We did not selectively report measures that were significant (note that we report both significant and non-significant results for the effect of the contact intervention in Tables 2 and 3).

In line with suggestions by Reviewer 3, we now don't consider the registration of our analysis a pre-registration as such because it was submitted after data collection was complete (see below for further elaboration on this point). Nonetheless, all analyses were conducted after data collection was complete. Our expectations for all variables are listed in the registration, however, we only registered a general analysis plan and relevant Mplus code as the analyses did not differ between variables.

The only measures not reported in the main manuscript are four scales which were measured at T1 and T3 only: experienced equality-based respect, perceived similarity between Leavers and Remainers, life satisfaction and well-being (see section A.3.5 in the Supplement). Life satisfaction and well-being outcomes are the focus of a separate paper exploring how Brexit and contact impact health. Equality-based respect and perceived similarity between Leavers and Remainers were measured in slightly different ways across timepoints, in relations to different aspects of contact (prior contact, the interaction, contact between T2 and T3). Moreover, several variables were assessed at T1 only (e.g., prior contact) – these were included in the attrition analyses and group comparisons.

- 3. The final sample size (N=120, so only ~60 per group) was relatively small, given the large pool of participants (N=484) available for the intervention. Why? There are obvious risks with running under-powered studies and this issue should be addressed.**

First, let us explain how the initial pool of participants was reduced to 120. Participants who indicated upon completion of the T1 questionnaire that they would like to participate in the interaction part of the study were contacted by one of the research assistants via the Prolific messaging system to indicate their availability for a number of available timeslots for the interaction. Of these participants, N=172 never responded. Of the N=310 participants who did respond, N=90 never confirmed a suitable slot for the interaction or, in a few cases, did not show for a scheduled interaction. N=120 completed the interaction. The remaining N=102 participants were Remainers who could not be assigned an interaction partner due to the fact that many more Remainers (N=394) than Leavers (N=192) entered the study to begin with (i.e.,

Dr Nicole Tausch
School of Psychology & Neuroscience
University of St Andrews
St Andrews, KY16 9JP
United Kingdom
Website
Google Scholar

there was a ‘surplus’ Remainers). We agree that this could have been clearer in the manuscript and now include more details on page 8:

“We obtained an initial sample of 586 respondents who completed the T1 (baseline) survey. Of these respondents, 484 (82.6%) indicated that they would like to continue with the study. These participants were contacted by the research assistants via the Prolific messaging system to indicate their availability for the available timeslots for the interaction. Of these participants, N=172 never responded and of the N=310 participants who did initially respond, N=90 never confirmed a timeslot or, in a few cases, did not show for a scheduled interaction. Finally, N=102 participants were Remainers who could not be assigned an interaction partner since many more Remainers (N=394) than Leavers (N=192) entered the study to begin with (see further details on participant recruitment in the Supplement).”

Thank you also for raising the important issue of power. Our final sample size was restricted due to both practical (the difficulty of scheduling interactions) as well as budgetary constraints (see further elaboration below). While 60 per condition might seem small and underpowered, please note that repeated measures design such as ours (i.e., the inclusion of baseline measures prior to the intervention) have much greater statistical power than a between-subjects only designs because as it allows controlling for factors that cause variability between subjects. If we consider our sample size of 120, a post-hoc sensitivity power analysis conducted in G*Power 3.1 for mixed designs with within-between interactions, with average correlations between repeated measures of outcome variables of $r_{T1-T2} = .545$ and $r_{T1-T3} = .607$, revealed that our sample size allows 80% power to detect an effect as small as $d = 0.246$ for tests of short-term effects and $d = 0.228$ for tests of long-term effects. A more conservative test, which considers dyads rather than individuals as the level of analysis ($N=60$), suggests that we have 80% power to detect effects of .350 and .326, respectively. We compared these detectable effect sizes to average effect size obtained in Pettigrew and Tropp’s (2006) meta-analysis, which was $d = .462$, with a 95% confidence interval of .451/.473 (Note that the meta-analysis also found a large overall effect of .713 for true experiments with random assignment). Given that the detectable effect size in our study at 80% power is smaller than the lower bound of the confidence interval reported in this meta-analysis, we believe that our study had sufficient power to detect a typical effect of intergroup contact and to test the effects of our contact intervention.

We now refer to the power of our experiment explicitly in the manuscript (p. 8):

Dr Nicole Tausch
School of Psychology & Neuroscience
University of St Andrews
St Andrews, KY16 9JP
United Kingdom
Website
Google Scholar

“Post-hoc sensitivity power analysis for mixed designs with within-between interactions, with average correlations between repeated measures of outcomes of $r_{T1-T2} = .545$ and $r_{T1-T3} = .607$ and considering dyads ($N=60$) as the unit of analysis revealed that our final sample size allows 80% power to detect effect sizes of $d = 0.350$ for tests of short-term effects and $d = 0.326$ for tests of long-term effects. Note that the detectable effects are smaller than the lower-bound of the confidence interval of overall contact effects reported in the largest meta-analysis of contact effects (Pettigrew & Tropp, 2006). We therefore believe that this sample size is sufficient to test our main hypotheses.”

4. Detailed comments [page#]:[para#]:[line#]

3:3:4 “has allows”  “allows”.

6:2:1 “asess”  “assess”.

14:Fig 3 There should be error bars on the points and the font size in caption and axes labels should be increased.

21:2:5 “contract”  “contact”.

26:4:2 “emotivational” ?

Thank you for pointing out the typographical mistakes; we have corrected them. We have also changed the term “emotivational” to “motivational” in the manuscript and supplement. The term emotivation stems from the emotion literature and refers emotion-specific goals (e.g., Roseman, 2008). However, given that this term is likely to be unknown to non-specialist audiences we opted for a more straightforward term which is also used in the literature. We have also changed Figure 3 so that it now presents raw data points:

Dr Nicole Tausch
School of Psychology & Neuroscience
University of St Andrews
St Andrews, KY16 9JP
United Kingdom
Website
Google Scholar

Roseman, I. J. (2008). Motivations and emotivations: Approach, avoidance, and other tendencies in motivated and emotional behavior. In A. J. Elliot (Ed.), *Handbook of approach and avoidance motivation* (pp. 343–366). Psychology Press

Reviewer #2

This is a well-written manuscript that makes a clear case for the need for more research, especially outside the US, on the effect of intergroup contact in the political domain on reducing affective polarization. The study uses a rigorous design with careful measurement to produce causal estimates of the impact of a well-informed intervention.

Thank you for this positive feedback!

- 1. However, the framing of the paper and discussion of the findings, in my view, oversell the results. First, the discussion claims that there are significant effects on more outcomes than appear to be the case given the results in the tables. But more importantly, the immediate effects of the intervention all decay by one month. I would suggest the authors update the tone and reframe the discussion to**

Dr Nicole Tausch
School of Psychology & Neuroscience
University of St Andrews
St Andrews, KY16 9JP
United Kingdom
Website
Google Scholar

be more grounded in these mixed findings and less optimistic about what they imply for our understanding of the power of intergroup contact in this domain.

We fully agree with this assessment and have thoroughly revised our interpretation and discussion of results throughout the paper. We now highlight only statistically significant interactions and refrain from referring near-significant interactions in the results section. We have also tempered our conclusions in the abstract and discuss these mixed findings more cautiously.

For example, the conclusion in the abstract now states:

“Our findings highlight the challenges of designing effective interventions that engender enduring attitude change in polarized contexts and of engaging those with extreme political views. This study can provide a useful framework for future research on this topic.”

In the discussion (p.22), we state:

“Overall, evidence for the success of our intervention can at best be considered mixed, as there were no statistically significant intervention effects for any of the other outcome variables, nor was there any evidence that these effects were sustained over the course of one month. This severely limits the direct applicability of our intervention and calls for more systematic research.”

I also have concerns about the power of the experiment and what we can conclude in terms of generalizability which I discuss more below.

- 2. First, the discussion section states that the intervention had significant effects on stereotypes, meta-beliefs and contempt. According to Tables 2 and 3, there is no effect on stereotypes or meta-beliefs and the effect on contempt (and on meta-intelligence) is only statistically significant at the 90% threshold. I am not sure on what basis the first claim is made, and I would want the authors to qualify the claims for which the statistical significance is weaker. The authors also state that the meta-stereotypes show a statistically significant reduction in the treatment group, but again, this is only statistically significant at 90%.**

Dr Nicole Tausch
School of Psychology & Neuroscience
University of St Andrews
St Andrews, KY16 9JP
United Kingdom
Website
Google Scholar

Thank you for pointing this out. We have corrected these mistakes and now only interpret findings at the 95% threshold as significant and refrain from highlighting near-significant results. We now state our criteria for considering a result as statistically significant more explicitly on page 12 in the section “Statistics and reproducibility”, where we state that we use a type I error rate of 0.05.

- 3. Second, the authors are otherwise transparent about the fact that there are no longer-term effects of the intervention. However, they still say the results are “highly encouraging” and that the results are of considerable importance for efforts to reduce affective polarization. Given the relative intensity of the intervention and the difficulty of recruiting participants, this very ephemeral impact does not, to me, clearly imply that governments or organizations should be investing in more of this work. I wanted the authors to justify their enthusiasm about these results in light of these challenges and temper the tone of the paper in describing just how impactful these findings are and what exactly they would imply for future policy.**

We fully agree and have adjusted the tone of our conclusions accordingly (see above). In the general discussion, we now highlight points that we believe can be informative for potential interventions and clearly highlight the challenges and limited evidence for the effectiveness of our intervention, in particular in the longer term.

- 4. I was struck but the relatively small sample size of the study. In a back of the envelope calculation, the authors are decently powered to detect effect sizes of about one-half of a standard deviation. Indeed, this is about the size of the effects on the outcomes they find are statistically significant. However, that is a relatively high bar for social science experiments. Typically, we see effect sizes of between 0.2 to 0.3 standard deviations. It could be then, that some of the estimated null effects are type II errors and due to the study being underpowered. I do not think this is likely the case for the effects estimated at T3 given that many are very close to 0; however, this could be the case for some of the other outcomes estimated at T2 where effects were found to be insignificant. This matters, in part, because of the promise of learning about not just whether but how intergroup contact works. One strength of the study is the thoughtful design of the intervention and outcome measures. This should help us understand whether this very particular**

Dr Nicole Tausch
School of Psychology & Neuroscience
University of St Andrews
St Andrews, KY16 9JP
United Kingdom
Website
Google Scholar

type of moderated discussion about political issues works, and on which outcomes. But because some of the outcomes may exhibit smaller effects, we may not know with any confidence whether the pattern of effects working on some outcomes but not others should be taken as instructive for our understanding of the mechanics of this intervention.

Thank you also for raising the important issue of power. Our final sample size was restricted due to both practical (the difficulty of scheduling interactions) as well as budgetary constraints (see further elaboration below). While 60 per condition might seem small and underpowered, please note that repeated measures design such as ours (i.e., the inclusion of baseline measures prior to the intervention) have much greater statistical power than a between-subjects only designs because as they allow controlling for factors that cause variability between subjects. If we consider our sample size of 120, a post-hoc sensitivity power analysis conducted in G*Power 3.1 for mixed designs with within-between interactions, with average correlations between repeated measures of outcome variables of $r_{T1-T2} = .545$ and $r_{T1-T3} = .607$, revealed that our sample size allows 80% power to detect an effect as small as $d = 0.246$ for tests of short-term effects and $d = 0.228$ for tests of long-term effects. A more conservative test, which considers dyads rather than individuals as the level of analysis ($N=60$), suggests that we have 80% power to detect effects of .350 and .326, respectively. We compared these detectable effect sizes to average effect size obtained in Pettigrew and Tropp's (2006) meta-analysis, which was $d = .462$, with a 95% confidence interval of .451/.473 (Note that the meta-analysis also found a large overall effect of .713 for true experiments with random assignment). Given that the detectable effect size in our study at 80% power is smaller than the lower bound of the confidence interval reported in this meta-analysis, we believe that our study had sufficient power to detect a typical effect of intergroup contact and to test the effects of our contact intervention.

We now refer to the power of our experiment explicitly in the manuscript (p. 8):

“Post-hoc sensitivity power analysis for mixed designs with within-between interactions, with average correlations between repeated measures of outcomes of $r_{T1-T2} = .545$ and $r_{T1-T3} = .607$ and considering dyads ($N=60$) as the unit of analysis revealed that our final sample size allows 80% power to detect effect sizes of $d = 0.350$ for tests of short-term effects and $d = 0.326$ for tests of long-term effects. Note that the detectable effects are smaller than the lower-bound of the confidence interval of overall contact effects reported in the largest meta-analysis of contact effects (Pettigrew & Tropp, 2006). We therefore believe that this sample size is sufficient to test our main hypotheses.”

Dr Nicole Tausch
School of Psychology & Neuroscience
University of St Andrews
St Andrews, KY16 9JP
United Kingdom
Website
Google Scholar

- 5. Related to power, it was unclear from the paper why there were only 120 participants who participated in the intervention when 484 indicated that they would like to have continued with the study. From the manuscript, it was unclear how the 484 participants got whittled down to 120 and why. After reading through the supplementary materials, I can see that the reasons include non-response or no-shows (with a minority of people not being able to be matched due to scheduling). The first two categories of reasons are especially important for thinking about generalizability. The authors nicely conduct tests for selective attrition across those who report willingness to continue in the study or not. But I also would have wanted to see this done for people who completed at least the intervention and T2 survey relative to those who dropped out.**

As requested, we now also include an additional test for selective attrition. Specifically, we compared participants who completed the interaction (N=120) with those who dropped out by not responding to our attempts to schedule an interaction or by not showing for a scheduled interaction (N=262) on all variables of interest. (As you noted, these two groups are the most important ones. We did not include participants who were responsive and willing to participate but could not be matched with an interaction partner in this analysis as non-participation was out of these participants' control. Note that our inability to find an interaction partner for these participants was due to the surplus of Remainders who entered the study.) These analyses yielded only two statistically significant effects and there were no significant interactions between completion and group (Leaver, Remainder). Specifically, we found that those who dropped out at this stage were younger and less willing to compromise with the outgroup, compared to those who completed the interaction. We can only speculate why these variables might have contributed to dropout at this stage. Participation in the interaction required participants to find a quiet space and period of uninterrupted time. It is plausible that younger participants might have more conflicting commitments and demands on their time, for example due to caring responsibilities for young children, which might have impacted on their ability to complete the study. Regarding the (lack of) willingness to compromise as a factor that may have led to dropout, we think it is possible that those who were less willing to compromise with the outgroup might have been put off by the ground rules for the interaction, which we laid out in our communications prior to the interaction. It is possible that participants who were less willing to compromise would have preferred a more confrontational interaction style and were less interested in engaging in dialogue.

Dr Nicole Tausch
School of Psychology & Neuroscience
University of St Andrews
St Andrews, KY16 9JP
United Kingdom
Website
Google Scholar

We have added detailed information on this addition attrition analysis in the supplement (pp. 21-23, Tables S6b and S7b) and added a forest plot for these analyses to Figure 2 in the manuscript and briefly describe these findings in the main text (p.13-14):

“We also compared participants who declared at T1 that they would like to participate but failed to respond or show (N= 262) with participants who participated in the interaction (N=120; see Figure 2, Panel b). The analyses indicated that younger people and those less willing to compromise with the outgroup were more likely to drop out at the later stages. There were no significant differences between Leavers and Remainers.”

Moreover, we also include more detail on how our initial sample got whittled down to 120 (p. 8). This now states:

“We obtained an initial sample of 586 respondents who completed the T1 (baseline) survey. Of these respondents, 484 (82.6%) indicated that they would like to continue with the study. These participants were contacted by the research assistants via the Prolific messaging system to indicate their availability for the available timeslots for the interaction. Of these participants, N=172 never responded and of the N=310 participants who did initially respond, N=90 never confirmed a timeslot or, in a few cases, did not show for a scheduled interaction. Finally, N=102 participants were Remainers who could not be assigned an interaction partner since many more Remainers (N=394) than Leavers (N=192) entered the study to begin with (see further details on participant recruitment in the Supplement).”

- 6. Finally, I liked the fact that the authors had a control condition that allows us to identify a very specific aspect of intergroup contact we care about: political interactions. However, it was unclear to me just how apolitical the control condition was. Were respondents in that condition aware of the fact that they were paired with an outgroup member or someone who had a different perspective about Brexit? Or were they aware that the goal of the study was to interrogate intergroup contact along this dimension? That would be important to know since I would expect that the findings would underestimate effects of contact if participants had some information about the intergroup nature of the exchange.**

We absolutely agree with you that this is an important point that needs to be considered explicitly. Prior to the interaction (i.e., in the participant information sheet provided at T1) participants were informed that: “The interaction will involve getting acquainted, discussing a

Dr Nicole Tausch
School of Psychology & Neuroscience
University of St Andrews
St Andrews, KY16 9JP
United Kingdom
Website
Google Scholar

current UK-wide issue (Brexit or the future of the Royal family), and finding possible solutions to a current social issue. Note that this may involve interacting with someone who holds an opposing view about Brexit or the Royal family.” (the information sheet is included in the materials for this study deposited on the OSF). So essentially, participants did not know in advance whether their interaction partner would be on the opposing side of the issue, nor did they know which issue they would be asked to discuss. Importantly, participants in the control group were at no point made aware of the fact that they were interacting with someone who had an opposing view on Brexit. Thus, the control condition does not entail *intergroup* contact along the Leaver-Remainer divide as such. While it is true that the control discussion is not entirely apolitical (it taps into issues of support for the monarchy, public funding and racism) and correlates with political opinions, ‘Megxit’ differs from ‘Brexit’ in that it is a less high-stakes issue which was outside of people’s political influence (i.e., there was no referendum on how it should be dealt with), Importantly, our control condition generated a good amount of engagement and difference in opinion to mimic the style of conversation in the experimental group, without making Brexit identities salient. All in all, we think that this makes for a strong control condition and a conservative test of the impact of intergroup contact. (see our response to Reviewer 3’s point 4 below, for further elaboration).

We now explicitly state what participants were told at the beginning of the study (p. 8):

“Participants were informed that the interaction would involve discussing a current UK-wide issue (Brexit or the future of the Royal family) and that it may involve interacting with someone who holds an opposing view on the issue.”

We have also included more information on the control group on page 9:

“Participants in the control group were asked to exchange their views on a controversial (though somewhat less politically charged) topic: the then recent departure of Prince Harry and Meghan Markle from the British royal family. This topic frequently featured in the media at the time and generated a good amount of engagement and difference in opinion to mimic the style of conversation in the experimental group. Importantly, although dyads in the control condition also consisted of a Leaver and a Remainer, participants in the control condition were not aware that they were interacting with someone with an opposing Brexit identity and this condition therefore did not constitute intergroup contact along the Leaver-Remainer divide⁴⁷. The choice of this control group ensured that any experimental effects could be attributed to engagement in intergroup contact rather than a positive interaction per se.”

Dr Nicole Tausch
School of Psychology & Neuroscience
University of St Andrews
St Andrews, KY16 9JP
United Kingdom
Website
Google Scholar

Reviewer #3

I am excited about the present research for several reasons. Even though there must, by now, be thousands of studies on the relationship between intergroup contact and prejudice, Paluck et al. (2018, 2020) only found 28 studies that fulfilled the strict criteria of a randomized controlled trial and of which only 8 studies measured outcomes at least one day after the contact-based intervention. As the present research does both of those things, it addresses the urgent need for evidence concerning the causal effects of intergroup contact. Further, the intervention stands out in its adherence to best practices informed by intergroup contact theory and other theoretical perspectives, in particular, its implementation of the sequential model of intergroup contact (Pettigrew, 1998). Another strength of the present research that is perhaps less obvious to researchers outside of the field of intergroup contact research is its control condition. Similar studies tend to rely on either a no-contact control group in which participants do not interact with anyone or an ingroup-contact control group in which participants interact with an ingroup member. The former is problematic as it does not allow isolating the effect of intergroup interaction from that of any social interaction while the latter is problematic as ingroup contact itself might have an effect on relevant outcomes. Tausch et al.'s choice of a control group is elegant as participants interact with an outgroup member in both condition but are only aware of the interaction partner's group membership in the experimental condition. For those reasons, I recommend the publication of this manuscript in *Communications Psychology*.

Thank you very much for these encouraging comments!

Still, I want to offer a few comments and recommendations.

1. First, I think that the authors overstate the support for the contact hypothesis in the literature at this moment. For example, I was surprised that the authors support a claim about the causal effects of intergroup contact—"intergroup contact robustly reduces negative intergroup attitudes"—with evidence from correlational studies (van Assche et al., 2023). In a sense, this framing undermines

Dr Nicole Tausch
School of Psychology & Neuroscience
University of St Andrews
St Andrews, KY16 9JP
United Kingdom
Website
Google Scholar

the importance of the present research which stands out as one of the rare experimental studies in this field. In any case, I think it is worth noting that recent research highlighted the limited experimental evidence (for a review, see Paluck et al. 2018, 2020) and contradicting longitudinal evidence (e.g., Friehs, Bracegirdle, et al., 2024, *Soc. Psychol. Pers. Sci.*; Hodson & Meleady, 2024, *Am. Psychol.*; Sengupta et al., 2023, *Am. Psychol.*; but see Gorska & Tausch, 2023, *Soc. Psychol. Pers. Sci.*) for the causal effects of intergroup contact.

Thank you for highlighting this, and for pointing us to this recent literature. To address your point, we have now re-written this paragraph (p.3):

“A vast literature on intergroup contact suggests that contact might be beneficial in reducing negative intergroup attitudes, especially when it takes place under conditions of equal status, involves cooperation in pursuit of a common goal, and is supported by institutional or informal norms^{23,24, 25}. Contact’s effects are thought to be enhanced to the extent that it allows participants to become personally acquainted²⁶ and encourages self-disclosure^{27,28} and generates positive affective ties to the outgroup²⁹. Nonetheless, much of the research underpinning contact theory has been correlational³⁰ and recent longitudinal work has yielded mixed findings regarding the within-person effects of contact^{31, 32, 33}. Importantly, there is a dearth of high-quality experimental intervention studies and the few studies available indicate that the effectiveness of contact interventions varies substantially according to the type of prejudice addressed³⁰.”

- 2. Second, the authors do not disclose that they registered the preregistration after they had collected the data (as stated in the preregistration). Readers might have different opinions on how useful preregistration (postregistration?) after data collection is so this fact should be disclosed in the main text. I, personally, do not see preregistration as particularly informative in most such cases.**

This is correct, and we apologize for not stating this more clearly in the previous version of the manuscript. We uploaded the grant proposal submitted in 2019 to the OSF (<https://osf.io/8rdps/>). This details the planned study design, some of the main constructs of interest, and outlines general hypotheses. However, it was unfortunately not feasible for us to pre-register the specific planned analyses and code prior to data collection. The study was prepared and conducted during the upheaval of the Covid lockdowns, which put extreme demands on the investigators’ time due to caring responsibilities, the switch to online teaching, and long-term illness. We prioritized data collection during this time and were only able to

Dr Nicole Tausch
School of Psychology & Neuroscience
University of St Andrews
St Andrews, KY16 9JP
United Kingdom
Website
Google Scholar

develop the specific analyses plan and code once this was completed. While we can assure you that no analyses were conducted prior to this registration, we accept that readers might find this less informative than a registration prior to data collection. In line with guidelines on the OSF, which would consider our procedure as a registration rather than pre-registration (<https://help.osf.io/article/550-registration-and-preregistration-faq-s>), we now refer to it as such. To address your concern, we state this explicitly on page 8:

“Although the study was not pre-registered, all analyses were registered and conducted after data collection was complete, and all outcome measures and conditions are reported.”

- 3. Third, the authors summarize the results from Table 2 as showing that “there were significant changes from T1 to T2 on most outcome variables in the expected directions in the intervention condition, and very few effects for the control group”. To me, this reads as somewhat misleading as most Time x Condition interactions are not significant. After all, the difference between “significant” and “not significant” is not itself statistically significant (<https://stat.columbia.edu/~gelman/research/published/signif4.pdf>).**

We fully agree with you and have changed our reporting of the results accordingly. We now clearly state what we consider a statistically significant effect of our intervention in the “Statistics and reproducibility” section (p. 12):

“We use a type I error rate of 0.05 and report two-tailed significance levels for all analyses. We employed multi-level analyses⁹⁷ using Mplus⁹⁸. We tested for experimental effects in a series of three-level models in Mplus, accounting for nesting of measurement points (level 1) within individuals (level 2) and nesting of individuals within dyads (level 3). In each of these models, time was coded with a dummy variable comparing measurement at T1 with measurement at T2 (to assess short-term effects) or measurement at T1 with measurement at T3 (to assess longer-term effects). Changes over time are represented by the effects of dummy variables capturing short-term (T1-T2) and longer-term (T1-T3) effects, respectively. Experimental effects are indicated by a significant cross-level interaction between condition (0 = control group, 1 = intervention group), which was specified as a level-3 variable, and the slope representing time.”

- 4. Lastly, I have a few clarification questions. Do you know whether Remainers and Leavers differed in their opinions on the control topic? In a sense, this is not an**

Dr Nicole Tausch
School of Psychology & Neuroscience
University of St Andrews
St Andrews, KY16 9JP
United Kingdom
Website
Google Scholar

unpolitical topic (as it touches on participants' attitudes concerning racism and the monarchy) and I would expect Leavers and Remainers to differ in their opinions on it.

Thank you for raising this important issue. Our rationale for choosing this particular control group, and how it differed from the experimental group, could certainly benefit from further clarification. When designing the control condition, we were looking for an issue that was topical and likely to be known to our participants, on which there were a range of different opinions, and which would allow us to design discussion questions that were structurally similar to those in the experimental condition. We were aiming to achieve a similar tone of, and engagement with, the discussion topic in the control and experimental groups. We considered a range of topics and settled for the Duke and Duchess of Sussex's departure from the royal family because it fulfilled these criteria. The Sussexes announced that they were stepping down from formal duties in January 2020 and formally relinquished their royal patronages in February 2021. This was followed by a number of controversies (e.g., around public funding of the couple's security detail) and accompanied by a large amount of media attention either criticizing or lauding the couple for their decision. It also generated heated debates about the future of the monarchy, the use of public funding to support the institution, and racism within the royal family. The term 'Megxit', a play on the term 'Brexit', was coined during this period. Our research commenced in November 2020 and interactions took place through to the end of April 2021, a period of time where this topic was of much public interest and frequently featured in media reports. While 'Megxit' differs from 'Brexit' in that it is a less high-stakes issue which was outside of people's political influence (i.e., there was no referendum on how it should be dealt with), it still generated a good amount of engagement and difference in opinion to mimic the style of conversation in the experimental group.

We agree with you that opinions on this topic (i.e., support for the couple, views on the monarchy) are not independent of political variables, and likely to be correlated with Leaver-Remainer identities. In fact, a recent survey revealed substantial differences of opinion between Leavers and Remainers about whether Prince Harry and Meghan Markle should be invited to the King's coronation, with 60 per cent of Remainers thinking that they should be invited, compared with 34 per cent of Leavers supporting this (see: <https://www.thetimes.com/uk/royal-family/article/harry-meghan-popularity-poll-latest-approval-rating-2023-wfnv6wmt>). Given that Leavers and Remainers differ on a number of dimensions (e.g., age; Curtice, 2017; Ford & Goodwin, 2017), this is not surprising.

Dr Nicole Tausch
School of Psychology & Neuroscience
University of St Andrews
St Andrews, KY16 9JP
United Kingdom
Website
Google Scholar

However, we believe that the fact that Leavers and Remainers likely differed in systematic ways in how they viewed the control issue is not a threat to the integrity of the design. Rather, we think it strengthens it because disagreement about the topic would have created a conversation that closely resembles that in the experimental group. The key difference between the control and experimental group is that, even though Leavers and Remainers were interacting with each other, Brexit identities were not at all mentioned during the interaction and participants were not aware of their interaction partner's Brexit identity. Thus, the control condition does not entail intergroup contact along the Leaver-Remainer divide as such (see Hewstone & Brown, 1986 on a discussion of the importance of category salience during intergroup contact). All in all, we think that this makes for a strong control condition and a conservative test of the impact of intergroup contact.

We now elaborate on the choice of control group on page 9:

“Participants in the control group were asked to exchange their views on a controversial (though somewhat less politically charged) topic: the then recent departure of Prince Harry and Meghan Markle from the British royal family. This topic frequently featured in the media at the time and generated a good amount of engagement and difference in opinion to mimic the style of conversation in the experimental group. Importantly, although dyads in the control condition also consisted of a Leaver and a Remainer, participants in the control condition were not aware that they were interacting with someone with an opposing Brexit identity and this condition therefore did not constitute intergroup contact along the Leaver-Remainer divide⁴⁷. The choice of this control group ensured that any experimental effects could be attributed to engagement in intergroup contact rather than a positive interaction per se.”

Curtice J (2016) The vote to leave the EU: Litmus test or lightning rod. In: Clery E, Curtice J, Harding R (eds) British Social Attitudes: The 34th Report (pp. 1–24). London, UK: Natcen Social Research.

Ford, R. & Goodwin, (2017). Britain After Brexit: A Nation Divided, Journal of Democracy, 28, 17–30.

Hewstone, M., & Brown, R. (1986). Contact is not enough: An intergroup perspective on the 'contact hypothesis.' In M. Hewstone & R. Brown (Eds.), Contact and conflict in intergroup encounters (pp. 1–44). Basil Blackwell.

Dr Nicole Tausch
School of Psychology & Neuroscience
University of St Andrews
St Andrews, KY16 9JP
United Kingdom
Website
Google Scholar

5. How were the participants selected from all those who agreed to participate in the interaction? How did you determine the sample size for this research?

Participants who indicated upon completion of the T1 questionnaire that they would like to participate in the interaction part of the study (N=484) were contacted by one of the research assistants via the Prolific messaging system to indicate their availability for a number of available timeslots for the interaction. Of these participants, N=172 never responded. Of the N=310 participants who did respond, N=90 never confirmed a suitable slot for the interaction or, in a few cases, did not show for a scheduled interaction. N=120 completed the interaction. The remaining N=102 participants were Remainers who could not be assigned an interaction partner due to the fact that many more Remainers (N=394) than Leavers (N=192) entered the study to begin with (i.e., there was a 'surplus' Remainers).

The sample size was determined by budgetary restrictions and practical issues that emerged during data collection, in particular the unforeseen difficulty of scheduling interactions. As is probably apparent from the previous paragraph, organizing the interactions was an incredibly time-intensive process which cost many hours of research assistance. The research was funded by a small grant (up to £10,000) where we initially planned to recruit 200 participants for this proof-of-concept study. We budgeted for appropriate re-imburement of participants to prevent dropouts across the three time-points and research assistance for scheduling and data collection. However, the difficulty of organizing interactions meant that our funds were depleted after collecting data from 120 participants and we were unable to conduct another recruitment drive.

We now mention how we determined the sample size on page 8:

"The final sample size was determined by practical and budgetary constraints."

We have also included more detail how we got to our final sample size on page 8:

"We obtained an initial sample of 586 respondents who completed the T1 (baseline) survey. Of these respondents, 484 (82.6%) indicated that they would like to continue with the study. These participants were contacted by the research assistants via the Prolific messaging system to indicate their availability for the available timeslots for the interaction. Of these participants, N=172 never responded and of the N=310 participants who did initially respond, N=90 never confirmed a timeslot or, in a few cases, did not show for a scheduled interaction. Finally, N=102 participants were Remainers who could not be assigned an interaction partner since many more

Dr Nicole Tausch
School of Psychology & Neuroscience
University of St Andrews
St Andrews, KY16 9JP
United Kingdom
Website
Google Scholar

Remainers (N=394) than Leavers (N=192) entered the study to begin with (see further details on participant recruitment in the Supplement)."

Again, we would like to thank you all for your encouraging and constructive feedback on our paper. We hope you like the revised version of our manuscript. It is much stronger thanks to your comments.

Yours sincerely,

Nicole Tausch, on behalf of all authors

29 June 2024

Dr Nicole Tausch
School of Psychology & Neuroscience
University of St Andrews
St Andrews, KY16 9JP
United Kingdom
Website
Google Scholar

Revision of “The Causal Effect of an Intergroup Contact Intervention on Affective Polarization around Brexit: A Randomized Controlled Trial” (MS COMMSPSYCHOL-24-0083-T)

Dear Reviewers,

We would like to thank you for your positive feedback on our revised manuscript!

Yours sincerely,

Nicole Tausch, on behalf of all authors

19 September 2024

Review of MS COMMSPSYCHOL-24-0083A

by Tausch et al.

Reviewer: Stephan Lewandowsky

Summary and Overall Recommendation

I reviewed this manuscript previously and my impressions at the time were already largely positive. I did, however, raise 3 major issues that prevented me from endorsing publication previously:

1. The authors seemingly exaggerated the Brexit divide in the UK relative to the partisan divide. In response, the authors correctly note that while the partisan divide had become more relevant (again) recently and is now dwarfing the Brexit divide, at the time that their data were collected the Brexit issue was still highly divisive. I accept the argument and the authors now explicitly state when the data were collected.
2. I was concerned by the number of different measures in the study which, given the absence of a strict preregistered analysis plan, might support inadvertent cherry-picking. The authors' response notes that they reported all results for all measures, including those that were nonsignificant, and that they no longer consider the “preregistration” to be a conventional preregistration. I accept the arguments.
3. The final sample size (N=120, so only ~60 per group) was relatively small, and I was concerned about the

Dr Nicole Tausch
School of Psychology & Neuroscience
University of St Andrews
St Andrews, KY16 9JP
United Kingdom
Website
Google Scholar

power of the study. In response, the authors conducted a thorough examination of the power issue and I believe that they have addressed this concern satisfactorily.
I have no further comments and I am happy to endorse publication of the paper.

Reviewer #2 (Remarks to the Author):

I am very satisfied with the authors' careful and comprehensive response to the initial set of reviews, both mine and from other reviewers. I support publication of this manuscript.

Reviewer #3 (Remarks to the Author):

I thank the author for engaging with my questions, comments, and suggestions. All my outstanding concerns have been addressed in the revision.